# Position: Significant impact of numerical precision in scientific machine learning

**Youngwoo Cho** [1] [*]   **Jaekak Yoo** [2] [*]   **Soyoung Yang** [1]   **Dong-Joon Yi** [3]
**Seung Mi Lee** [4]   **Mun Seok Jeong** [5]   **Jaegul Choo** [1]

## Abstract

The machine learning community has focused on computational efficiency, often leveraging lower-precision formats such as FP16, rather than the standard FP32. In contrast, little attention has been paid to higher-precision formats, such as FP64, despite their critical role in scientific domains like materials science, where even small numerical differences can lead to significant inaccuracies in physicochemical properties. This need for high precision extends to the emerging field of *machine learning for scientific tasks*, yet it has not been thoroughly investigated. According to several studies and our experiments, models trained with FP32 and FP64 can yield different scientific conclusions, yet this discrepancy is currently underreported, indicating that numerical precision is also a critical factor in scientific machine learning, as in traditional scientific computing. This precision issue limits the potential of scientific machine learning to serve as a reliable alternative or complement to traditional scientific computing in practical research. Our position paper not only highlights these precision-related issues but also recommends reporting comparisons between FP32 and FP64 results, encouraging the release of FP64 models. We believe that these efforts can enable machine learning to contribute meaningfully to the natural sciences, ensuring both scientific reliability and practical applicability.

## 1. Introduction

The rapid advancements in natural language processing (NLP) and computer vision (CV) in the field of machine learning (ML) have accelerated the broad application across various domains (Litjens et al., 2017; Ozbayoglu et al., 2020; Raghu & Schmidt, 2020; Ren et al., 2021; Lai et al., 2024). Specifically, *scientific ML*, which has begun to resolve intellectually demanding problems in scientific fields, has been highlighted across disciplines, opening new possibilities for scientific breakthroughs. In recognition of these advances, the 2024 Nobel Prize in Chemistry honored the contributions of scientific ML, highlighting innovations such as AlphaFold and RoseTTAFold (Jumper et al., 2021; Baek et al., 2021; The Royal Swedish Academy of Sciences, 2024). These models transformed scientific research by rapidly delivering results that once required significant resources and time-consuming experiments or simulations. Building on these successes, scientific ML not only addresses traditional labor-intensive workflows but also finds hidden patterns within complex data, thereby providing human researchers with direct insights into novel discoveries across natural sciences (Webb et al., 2018; Morgan & Jacobs, 2020; Karagiorgi et al., 2022; Yoo et al., 2023; 2024; Lu et al., 2024; Suh et al., 2025).

In the context of methodology, the development of scientific ML naturally follows the broader trends and paradigms of the ML research field. In the early stages of NLP and CV, most work focused on discriminative tasks (*e.g.*, named entity recognition and image classification) (Walker et al., 2006; Deng et al., 2009) before gradually shifting to generative tasks (*e.g.*, machine translation and text-to-image generation) (Bojar et al., 2014; Schuhmann et al., 2022). Further, generative approaches have advanced sequentially, moving from variational autoencoders (VAEs) to generative adversarial networks (GANs), and more recently, to diffusion models (Kingma & Welling, 2014; Goodfellow et al., 2014; Song & Ermon, 2019; Ho et al., 2020). In a similar manner, numerous scientific domains have rapidly adopted the latest advances from the ML community. For example, among various areas of bioinformatics, research on DNA sequence data initially leveraged discriminative models such as DeepVariant (Poplin et al., 2018) and DeepSEA (Zhou & Troyanskaya, 2015), and over time, the field shifted toward

---

[*]Equal contribution  [1]Kim Jaechul Graduate School of Artificial Intelligence, Korea Advanced Institute of Science and Technology, Daejeon, South Korea [2]Department of Chemistry, University of Illinois at Urbana-Champaign, Urbana, Illinois, USA [3]Department of Electronic Engineering, Hanyang University, Seoul, South Korea [4]Korea Research Institute of Standards and Science, Daejeon, South Korea [5]Department of Physics, Hanyang University, Seoul, South Korea. Correspondence to: Mun Seok Jeong <mjeong@hanyang.ac.kr>, Jaegul Choo <jchoo@kaist.ac.kr>.

*Proceedings of the 43$^{rd}$ International Conference on Machine Learning*, Seoul, South Korea. PMLR 306, 2026. Copyright 2026 by the author(s).

generative models including ExpressionGAN (Zrimec et al., 2022) and Evo (Nguyen et al., 2024). Similarly, material structure prediction in the field of materials and drug discovery has followed this trend from VAEs (Sanchez-Lengeling & Aspuru-Guzik, 2018; Gómez-Bombarelli et al., 2018; Lim et al., 2018) and GANs (Prykhodko et al., 2019; Kim et al., 2020; Abbasi et al., 2022) to diffusion models (Hoogeboom et al., 2022; Peng et al., 2023; Zeni et al., 2025).

In parallel with these advances, the scaling law, one of the most recent paradigms in ML research, emphasizes performance enhancement by progressively increasing the size of models, training datasets, and computational resources (Kaplan et al., 2020; Snell et al., 2025). Building upon this idea of incremental scale expansion, researchers have successfully tested the approach of *bigger is better* across diverse fields, including NLP, CV, reinforcement learning, and time-series forecasting (Zhai et al., 2022; Cherti et al., 2023; Hilton et al., 2023; Neumann & Gros, 2023; Shi et al., 2024). Following this pattern, scientific ML is also adopting this paradigm, and in fact, large models designed to address scientific tasks have already begun to appear (Nguyen et al., 2024; Zhang et al., 2024; Wood et al., 2025).

As these models grow larger and more complex, they require massive computational resources, presenting significant challenges for both training and inference processes. To address this, lower numerical precision and quantization are widely employed strategies that help reduce computational cost (Micikevicius et al., 2018; Zhu et al., 2024). These approaches inevitably involve a trade-off between fidelity and resource efficiency, typically resulting in some accuracy degradation. To minimize such precision-related losses, techniques such as mixed precision training (Micikevicius et al., 2018) and sophisticated quantization methods (Banner et al., 2019; Dettmers et al., 2022; Liu et al., 2023; Xu et al., 2024) have been proposed, which allow researchers to preserve the original accuracy while achieving the advantages of reduced computational costs. Consequently, the ML community has accepted slight accuracy degradation as a natural trade-off for greater efficiency, thereby integrating these lower-precision techniques into real-world applications to balance performance and computational burden.

However, the tolerance for lower-precision techniques raises substantial concerns in the field of scientific computing. Scientific computing primarily aims to solve fundamental physics equations that are difficult to solve manually by simplifying or discretizing the continuous and infinite real-world phenomena to make them computationally tractable. As a consequence, even subtle differences in numerical precision can lead to significant issues regarding the reliability of computational results. Our experimental findings demonstrate that *single-precision arithmetic introduces numerical errors that substantially affect the accuracy of computed so-*lutions to fundamental physical equations. These errors can cause substantial deviations in computed quantities, including molecular energies, forces, and optimized geometries, vibrational frequencies that alter mode ordering and peak assignment, and electromagnetic transmittance spectra and nonlinear harmonic signals, thereby reducing the reliability of the results. Importantly, these precision-related challenges are not confined to traditional computational science, as ML models are increasingly utilized in various studies to replace conventional simulations. Since traditional computational science requires high precision, it becomes essential to verify whether ML models trained or evaluated in single precision produce scientifically valid results.

**In this position paper, we argue for the significant role of numerical precision in scientific ML research, emphasizing the need for evaluating and analyzing its impact on results derived from varying precision levels.** To this end, we first highlight real-world examples from established computational simulations in which numerical precision directly affects the results. We then explain that the importance of numerical precision is not confined to traditional scientific computing alone but is also deeply related to ML applications in scientific domains. Specifically, we provide examples involving ML potential models and physics-informed neural networks (PINNs), which are actively studied in both ML and science domains, demonstrating the critical role of numerical precision in these areas (Raissi et al., 2019; Kocer et al., 2022; Käser et al., 2023). Additionally, we explore the implications of large language models (LLMs) in scientific ML on precision-related considerations.

In conclusion, we present concrete recommendations for the ML community and potential research directions based on our discussions. We then provide alternative viewpoints to our position, offer responses, and conclude. Since the main role of ML in scientific research is to deepen understanding in traditional domains, the issues we raise must be rigorously examined. When relatively simple actions by ML researchers can remove barriers that hinder natural scientists from applying ML models, these measures become essential, not optional. As scientific ML is still in its early stages, we hope that thorough debate will help minimize trial-and-error in future research.

## 2. Importance of numerical precision in scientific computing

The main goal of scientific computing is to solve complex physics equations through computational power, especially when manual solutions are impractical or nonexistent. Specifically, many-body problems including multiple object interactions demonstrate the necessity of high-performance computing. Accordingly, various computational methods have emerged to solve fundamental physics

equations: molecular dynamics for Newton's Second Law, density functional theory (DFT) (Jones & Gunnarsson, 1989) for the Schrödinger equation, and the finite-difference time-domain (FDTD) (Yee, 1966) method for Maxwell's equations. Despite the algorithmic progress outlined above, the fidelity of these simulations is bounded by how continuous physical variables are encoded on digital hardware. Modern digital processors represent real numbers as finite-length bit strings, so that continuous equations–ranging from $F = ma$ to the Schrödinger and Maxwell formulations–cannot be solved exactly. To bridge this gap, scientists adopt controllable approximations: reformulating the problem (*e.g.*, the Kohn–Sham equation (Kohn & Sham, 1965)) or discretizing time and space (*e.g.*, molecular dynamics). These methods remain reliable only when round-off error is tightly bounded, making double-precision arithmetic the *de facto* standard for balancing cost and accuracy. For instance, Quantum ESPRESSO (Giannozzi et al., 2009), a leading open-source DFT implementation, strictly enforces double precision throughout its code.

To empirically demonstrate the crucial role of precision in scientific computing, we present concrete examples showing how small numerical variations can significantly affect computational results, and analyze their impact in realistic research scenarios. Specifically, we demonstrate these effects through materials research scenarios, examining their implications and highlighting challenges arising from numerical accuracy.

### 2.1. Impact on density functional theory simulation

Quantum mechanics, beginning with Planck's quantum hypothesis (Planck, 1900), revolutionized our understanding of microscopic phenomena. While exact calculations are only possible for simple systems like the hydrogen atom, the Kohn-Sham equation introduced DFT as an efficient approach for many-body electron problems. Using the PySCF library (Sun et al., 2018), we performed geometry optimization calculations for water ($H_2O$) using both Hartree-Fock (HF) and DFT calculations with B3LYP functional and 6-311++G(d,p) basis set (Becke, 1993; Yanai et al., 2004; Andersson & Uvdal, 2005; Tirado-Rives & Jorgensen, 2008).

Figure 1 (a) shows the results of geometry-optimized water molecules obtained from HF and DFT calculations under FP32 and FP64 numerical precision conditions. When utilizing FP64, both HF and DFT calculations successfully converged within three optimization steps while satisfying the convergence criteria. Since DFT explicitly accounts for electron correlation effects (Becke, 1988), it is generally expected to provide more accurate results than HF, a trend that is also reflected in our findings. Comparing bond lengths, the reference (Bowen & Sutton, 1958) O-H bond length is 0.957 Å, while HF exhibits a deviation of 0.016 Å

(1.7 % error), and DFT yields a smaller deviation of 0.005 Å (0.5 % error). Similarly, for the bond angle, HF deviates by $1.7°$ (0.7 % error) from the reference value of $104.52°$, whereas DFT shows a smaller deviation of $0.55°$ (0.5 % error). However, when using FP32, significant numerical instabilities arise, preventing the convergence of optimization steps. In the case of HF calculations, the gradient of hydrogen atoms stagnates between 0.2–0.4 Ha/Bohr, which is significantly above the desired convergence threshold of $10^{-6}$ Ha/Bohr. For DFT calculations, the issue becomes even more pronounced, as the gradient values rapidly diverge beyond $10^5$ Ha/Bohr, resulting in termination before reaching the maximum step. As a result, when using FP32, the HF calculation exhibits a substantial 50 % error, while the DFT calculation produces a molecular structure that cannot exist in reality, as illustrated in Figure 1 (a). A detailed examination is provided in Appendix B.1.

### 2.2. Impact on finite difference time domain simulation

Electromagnetism, established by Maxwell's equations, provides the theoretical foundation for understanding electromagnetic waves. To address the computational challenges of solving Maxwell's equations, FDTD discretizes them in time and space. Using Meep (Oskooi et al., 2010), an open-source FDTD software, we investigated numerical precision effects on electromagnetic simulations, comparing FP32 and FP64 in nonlinear Kerr media simulations. We simulated a Kerr medium with a refractive index of 1.65, excited by an electromagnetic wave source ($\lambda = 1.55 \, \mu m$, $\Delta\lambda = 0.15 \, \mu m$).

Figure 1 (b) presents the transmission spectrum of the nonlinear Kerr medium under FP32 and FP64 precision settings. From left to right, the spectral peaks correspond to the fundamental generation induced by the source, the second harmonic generation (SHG), and the third harmonic generation (THG). While the fundamental peak exhibits minimal differences between FP32 and FP64, notable discrepancies arise in the SHG and THG regions. Specifically, FP32 calculations display pronounced background signal instability and intensity variations in harmonic generation, which result from imprecise numerical computation. A particularly notable difference appears in the behavior of the background signal. In FP64 calculations, the background follows a well-defined periodic pattern governed by the electromagnetic wave, whereas in FP32, the background signal appears as unstructured Gaussian-like noise. This phenomenon indicates that the lack of numerical precision in FP32 significantly disrupts the accurate computation of low-intensity transmitted power, particularly for electromagnetic waves in the range of $10^{-11}$ W/m$^2$. These findings highlight the fundamental limitations of single precision in reliably capturing weak electromagnetic signals and nonlinear optical effects.

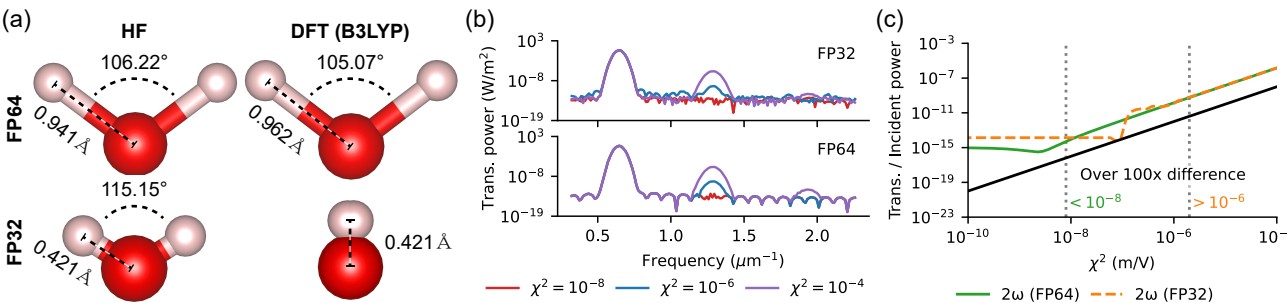

*Figure 1.* (a) Geometry optimizations of a water molecule using FP64 (top) and FP32 (bottom) with HF (left) and DFT (right) methods. FP64 computations yield physically valid structures, whereas FP32 leads to unrealistic geometries. (b) Transmittance spectra comparison between FP32 (top) and FP64 (bottom) in Kerr media, showing FP32's failure to accurately model higher harmonics and low-power wave patterns below $10^{-10}$ W/m$^2$. (c) Computed second harmonic susceptibility shown in FP64 (green solid) and FP32 lines (orange dashed) compared to theoretical quadratic behavior (black). FP64 maintains accuracy to $10^{-8}$ m/V, while FP32 deviates above $10^{-6}$ m/V, making it unsuitable for typical nonlinear materials.

To further analyze the impact of numerical precision, we examined the relationship between second-order nonlinear susceptibility ($\chi^2$) and the transmittance-to-incident power ratio. As shown in Figure 1 (c), the black upward-sloping line represents a quadratic line, serving as a reference line indicating the expected computational trend of transmittance over incident power ratio as nonlinear susceptibility varies. Ideally, the computationally simulated values should align with this reference trend, maintaining the same slope. Comparing the results obtained from FP64 (green solid line) and FP32 (orange dashed line), we observe that as nonlinear susceptibility decreases beyond a certain threshold, the ratio begins to saturate. This saturation point effectively defines the lower bound of computational precision achievable under each numerical setting.

Specifically, for values of $\chi^2$ above $10^{-6}$, both FP64 and FP32 provide reliable computational precision. However, for values below this threshold, FP32 results begin to exhibit saturation, rendering further calculations meaningless due to the loss of numerical resolution. In contrast, FP64 maintains simulation accuracy down to approximately $10^{-8}$, demonstrating a computational precision that is at least two orders of magnitude higher than that of FP32. This result implies that for most nonlinear materials with $\chi^2$ values below $10^{-6}$, transmittance spectrum simulations using FP32 become inherently unreliable. These findings highlight the critical role of numerical precision in computational science, particularly in fields where small numerical deviations can lead to substantial errors.

As demonstrated in both DFT and FDTD simulations, the limitations of single precision introduce significant inaccuracies, especially in cases involving highly sensitive physical properties. This highlights the necessity of carefully selecting numerical precision levels when conducting computational simulations, where maintaining the reliability of results is essential. While FP32 is adequate for various rou-

tine or weakly nonlinear calculations, our benchmarks show that in strongly nonlinear regimes it can introduce critical artifacts; the exact thresholds and representative case studies are provided in Appendix B.2.

## 3. Numerical precision issue in scientific ML

As demonstrated in the previous section, numerical precision is a critical factor in traditional scientific simulations, where even small numerical deviations can invalidate physical conclusions. This raises a fundamental question for scientific ML: **Do ML models used for scientific tasks inherit similar precision sensitivities?** We investigate this through case studies and experiments that differ in how strongly they are supported. ML potentials provide controlled experimental evidence (Section 3.1), PINNs draw on prior findings in the literature (Section 3.2), and LLMs raise a forward-looking concern (Section 3.3).

### 3.1. Impact on machine learning potentials

The first example we present is ML potentials[1] (Kocer et al., 2022; Käser et al., 2023), which is closely related to the DFT calculations discussed in Section 2.1. ML potential models compute potential energies and associated forces for material structures, offering faster alternatives to traditional quantum mechanical calculations (Pukrittayakamee et al., 2009; Smith et al., 2017; Gilmer et al., 2017; Schütt et al., 2017). Currently, many of these models (Batatia et al., 2022; Batzner et al., 2022; Deng et al., 2023; Park et al., 2024b; Fu et al., 2025; Wood et al., 2025) are integrated into widely used simulation packages such as ASE (Bahn & Jacobsen, 2002; Larsen et al., 2017) and LAMMPS (Plimpton, 1995; Thompson et al., 2022). Given this widespread adoption in practical research, numerical precision issues in these models could significantly impact scientific discoveries.

---

[1]The term *machine learning interatomic potential* is also used.

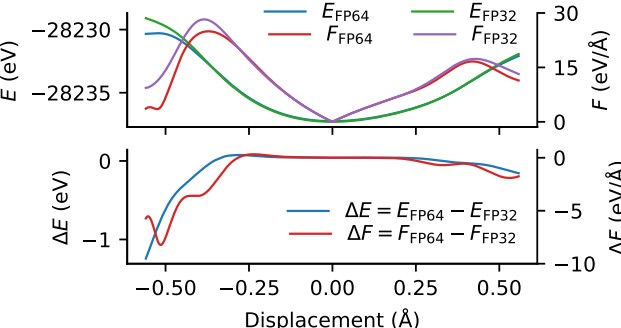

*Figure 2.* One-dimensional scan of the PES of the oseltamivir molecule obtained by displacing a selected oxygen atom along a specific direction, as illustrated in Figure 5. The top panel shows energy and force predictions from models trained with FP32 and FP64 precision as functions of the displacement relative to the initial structure. The bottom panel shows the corresponding differences in energy and force predictions between FP64 and FP32 models, which are negligible near the initial structure but increase with larger displacements.

To investigate this, we selected MACE (Batatia et al., 2022; Kovács et al., 2025), a widely used and representative ML potential, and trained the model from scratch in both FP32 and FP64 precision to isolate the impact of numerical precision on model behavior (training details are provided in Appendix B.3). Table 5 in Appendix B.3 reports the energy and force mean absolute errors (MAEs) of both models on the training and test sets. The FP64 and FP32 models show comparable MAE across different splits, with differences that appear negligible at first glance.

However, comparable errors do not necessarily imply that the two models behave similarly in practice. To illustrate this, we analyzed how precision affects the potential energy surface (PES) of *oseltamivir*, the active ingredient of the anti-influenza drug *Tamiflu*. We displaced one oxygen atom along a specific path (see Figure 5) and compared the predicted energies and forces. As shown in Figure 2, the discrepancies between the FP32 and FP64 models increase substantially with displacement, reaching up to 1 eV for energy and 10 eV/Å for force, respectively. Such large precision-dependent discrepancies are particularly concerning for applications involving non-equilibrium structures, such as reaction pathways or transition state searches.

Near the optimal structure, the differences appear modest: 41 meV (0.82 meV/atom) for energy and 6.6 meV/Å for force. Nevertheless, Figure 4 shows that structure optimizations using FP32 and FP64 converge to slightly different equilibrium geometries. While the overall conformations appear visually similar, subtle differences in atomic positions are evident. Although such differences may seem negligible, even minor geometric variations can significantly affect derived properties in quantum mechanical calculations.

*Table 1.* Selected vibrational mode frequencies (cm$^{-1}$) of the Oseltamivir molecule computed using the `MACE-OFF23-medium` models trained in FP32 and FP64 precision (complete results in Table 6). $\Delta$ denotes the absolute frequency difference between FP32 and FP64 models.

| GPU | Mode | Frequency (cm$^{-1}$) | | |
| --- | --- | --- | --- | --- |
| | | FP32 | FP64 | $\Delta$ |
| | 114 | 1496.89 | 1497.90 | 1.01 |
| H200 | 115 | 1499.69 | **1502.75** | 3.06 |
| | 116 | **1502.53** | 1507.71 | 5.18 |
| | 114 | 1496.85 | 1497.90 | 1.05 |
| B200 | 115 | 1499.74 | **1502.75** | 3.01 |
| | 116 | **1502.58** | 1507.71 | 5.13 |

*Table 2.* Vibrational mode misalignments between FP32 and FP64 `MACE-OFF23` models, grouped by molecular size. "Affected molecules" counts molecules with at least one misalignment; "Misalignments" counts the total across all modes.

| Atom count | Molecules | Affected molecules | Misalignments |
| --- | --- | --- | --- |
| 1-10 | 3,824 | 1,285 (33.6 %) | 2,208 |
| 11-20 | 10,882 | 7,121 (65.4 %) | 17,121 |
| 21-50 | 34,784 | 33,686 (96.8 %) | 228,616 |
| 51+ | 314 | 314 (100.0 %) | 7,688 |

To demonstrate this, we computed the vibrational frequencies of the relaxed structures obtained from both models (a complete list is provided in Table 6). Table 1 highlights selected modes for which FP32 and FP64 predictions differ by up to 5.2 cm$^{-1}$. While these differences appear small in absolute terms, vibrational spectroscopy relies on frequency shifts and mode ordering for peak assignment rather than absolute values. As a concrete example, a vibrational peak observed at 1502 cm$^{-1}$ would be assigned to different vibrational modes depending on whether FP32 or FP64 predictions are used (modes 115 and 116). Since each vibrational mode corresponds to distinct atomic displacement patterns and symmetry properties, this precision-dependent misassignment leads to fundamentally different physical interpretations. Critically, these discrepancies (1–5 cm$^{-1}$) fall within the spectral resolution of modern Raman and infrared spectroscopy, which routinely reaches the few-cm$^{-1}$ to sub-cm$^{-1}$ range. Experimental data obtained with such high-resolution instrumentation should not be undermined by numerical precision artifacts in the computational predictions used for spectral assignment.

The oseltamivir case above is a single illustrative example, and a natural question is whether such misassignments are incidental or systematic. To address this, we extended the analysis to the entire `MACE-OFF23` test set (Kovács et al., 2025), where 49,804 of 50,195 molecules were successfully computed. Here, we identify a mode misalignment as

a case where the FP64 mode $i$ with frequency $f_i^{\mathrm{FP64}}$ and its nearest-frequency FP32 mode $j$ with frequency $f_j^{\mathrm{FP32}}$ satisfy $i \neq j$ and $|f_i^{\mathrm{FP64}} - f_j^{\mathrm{FP32}}| < 1 \text{ cm}^{-1}$. To avoid near-degenerate false positives, we exclude FP64 modes within $0.5 \text{ cm}^{-1}$ of each other, so the reported counts are conservative. As shown in Table 2, 42,406 of the 49,804 successfully computed molecules (85.15 %) exhibit at least one misalignment, averaging 6.03 per affected molecule, and the affected fraction rises with molecular size. The point is not which precision is correct, but that precision alone reorders mode assignments across most of a standard benchmark. Since these assignments determine how experimental spectra are interpreted, this is precisely the kind of underreported issue our position highlights.

More broadly, this issue extends beyond vibrational spectroscopy. Recent ML potentials increasingly predict quantities derived directly from DFT calculations, such as Hamiltonians and electronic densities (Li et al., 2022; Yu et al., 2023; Wang et al., 2024), which are intrinsically more sensitive to numerical errors. The vibrational misassignment demonstrated here thus represents a broader reliability challenge in ML potential research.

### 3.2. Impact on physics-informed neural networks

Beyond the fundamental equations mentioned in the previous section, various subfields of natural science describe natural phenomena using differential equations. For example, in fluid dynamics, including weather prediction, Navier-Stokes, continuity, and heat transfer equations are used (Tritton, 2012; Bauer et al., 2015). Moreover, differential equations such as the Black-Scholes equation (Black & Scholes, 1973) are also employed in fields beyond natural sciences, such as financial engineering. Many of these equations either lack general analytical solutions or are too complex to be solved manually. As a result, numerical methods have been developed over time, leading to techniques such as the Euler method, Runge-Kutta methods, and Picard method (Strauss, 2007; Butcher, 2016). These techniques have also influenced modern approaches in ML, including diffusion models, NeuralODEs, and deep equilibrium models (DEQs) (Chen et al., 2018; Bai et al., 2019; Song & Ermon, 2019; Ho et al., 2020).

The concept of PINNs (Raissi et al., 2019) leverages automatic differentiation (autograd), fundamental to backpropagation, to solve differential equations using neural networks. Due to its simple yet powerful approach, PINNs have been widely adopted in scientific domains that rely on numerical methods. This section explores whether numerical precision issues also arise in PINNs and investigates related challenges through a literature survey.

First, Nakamura et al. (2022) explicitly discussed the impact of numerical precision in scientific research, reporting that training PINNs with FP32 failed, whereas FP64 did not: *from a comprehensive standpoint, FP32 computation has a risk of failure for the present problem compared with FP64.* This work applies PINNs to a fluid dynamics problem involving surface tension modeling, governed by a fourth-order differential equation. Notably, the original work attributes this failure to the loss of significant digits during the optimizer's line search, rather than to the higher-order automatic differentiation itself, illustrating that precision can affect scientific ML training through the optimization process. Although specific, this is a real-world scientific study, demonstrating that precision issues can significantly impact the practical use of PINNs.

Meanwhile, Sharma & Shankar (2022) explicitly accounted for precision issues and leveraged this understanding to improve the methodology of PINNs. The key idea of their work is to replace certain autograd operations in PINNs with a specialized finite difference method, reducing the computational cost associated with autograd. Here, to compensate for the loss of accuracy introduced by finite difference approximations, the authors proposed using high-precision (FP64) training. As a result, the reduction in computational cost from bypassing autograd exceeds the overhead introduced by FP64 operations, leading to an overall speedup that makes their approach faster than a vanilla PINN in FP32. Beyond the fields of PINNs and scientific ML, this study introduces a novel perspective on utilizing high-precision models in neural network research.

Thus, in the context of PINNs, a comprehensive study is needed to systematically assess the impact of numerical precision issues on scientific research. Fortunately, many fields share similar types of differential equations, *e.g.*, Laplace equation in electrostatics and fluid dynamics, where it describes electric potential distribution and velocity potential in inviscid flow, respectively. By focusing on the precision challenges of commonly used differential equations and rigorously validating PINNs in this context, such research could have a substantial impact across multiple domains.

### 3.3. Emerging concerns for large language models

The emergence of LLMs in scientific applications is accelerating, further raising concerns about numerical precision in such domains. Unlike the previous sections, which are grounded in controlled experiments or established literature, this section is intentionally forward-looking: as LLMs are increasingly applied to scientific tasks, we flag precision-related risks that we expect to grow, examining existing studies that point to such concerns rather than presenting settled experimental findings of our own.

The integration of LLMs in scientific domains follows two distinct approaches. The first involves direct inference without architectural modifications, where scientific data is trans-

formed into natural language format for existing LLM architectures (Jacobs et al., 2024; Niyongabo Rubungo et al., 2025; Liu et al., 2025). The second approach develops specialized architectures that combine domain-specific encoders with fine-tuned language models, preserving the intrinsic properties of scientific data while leveraging LLM capabilities (Li et al., 2024; Park et al., 2024a).

Regarding the first approach, recent research (Yuan et al., 2025) identifies significant fragility in the reproducibility of LLM inference. Even when utilizing deterministic greedy decoding, model outputs can diverge significantly due to variations in system configurations, such as evaluation batch size, GPU count, hardware versions, and numerical precision. Across multiple model families, Yuan et al. (2025) show that these factors all stem from a single cause: the non-associativity of floating-point arithmetic, where $(a + b) + c \neq a + (b + c)$, under finite numerical precision. In complex reasoning models, minor rounding discrepancies in early tokens can cascade into entirely different chains of thought, leading to accuracy variations of up to 9% and substantial fluctuations in response length. This numerical nondeterminism suggests that LLM-based scientific reasoning is highly sensitive to the underlying hardware execution order, making exact reproduction across different environments a critical challenge.

Another critical consideration in LLM deployment is the continuous increase in model size. For instance, the open-source Llama series demonstrates this trend clearly: LLaMA (65B parameters) grew to Llama-2 (70B) and further to Llama-3.1 (405B) (Touvron et al., 2023a;b; Llama Team, 2024), and more recently, DeepSeek-v3 has pushed this expansion even further, reaching 671B (DeepSeek-AI, 2024). Such explosive growth in model sizes across LLMs has resulted in a substantial increase in computational costs for both training and inference. To mitigate these costs, researchers commonly employ parameter quantization techniques by reducing model precision to lower-bit formats (Liu et al., 2021; Dettmers et al., 2022; Liu et al., 2023), sometimes even 1-bit representations (Xu et al., 2024).

However, these optimization strategies fundamentally conflict with the stringent precision requirements of scientific computing applications, as emphasized throughout our analysis. This issue is particularly critical for the second approach, where domain-specific encoders, which are often derived from scientific ML models, serve as feature extractors. If quantization significantly reduces the precision of the extracted features, the LLMs may fail to process them accurately, potentially degrading overall model performance. For example, Li et al. (2024) employed UniMol (Zhou et al., 2023), a model broadly categorized as an ML potential, as an encoder. Even if the encoder provides highly precise features, the LLM's lower precision representations may

obscure this information, leading to inaccurate final predictions. This inherent trade-off between computational efficiency and numerical precision highlights the necessity of careful consideration when integrating LLMs into scientific applications.

## 4. Call to Action

Building upon previous discussions, we present key suggestions for the scientific ML community.

**Benchmarking and reporting FP32 vs. FP64 results.**
Scientific ML models can be more sensitive to numerical precision than general ML tasks, where the choice between FP32 and FP64 may lead to different scientific conclusions. At the same time, higher precision comes at a cost, as factors such as training time, inference latency, and energy consumption remain significant constraints. Consequently, researchers should explicitly report the numerical precision used in their studies, conduct comparative analyses between FP32 and FP64 where applicable, and publicly release FP64-trained models to improve reproducibility and facilitate collaborative research. As a practical minimum standard when full FP32/FP64 retraining is infeasible, researchers can fix the hyperparameters tuned in one precision setting, perform a single additional run in the other, and compare only the precision-sensitive derived properties (*e.g.*, vibrational frequencies), briefly reporting the outcome in an appendix.

To support meaningful evaluations, standardized benchmarks that capture precision sensitivity across diverse scientific tasks are essential. For instance, benchmarks may include standard tasks such as bulk modulus calculations in DFT or standard electromagnetic simulations in FDTD, which are simple in formulation while remaining highly sensitive to numerical precision. In designing such benchmarks, our case studies suggest a useful starting point: precision sensitivity tended to appear more often for quantities involving high-order derivatives, strong nonlinearity, or derived properties (*e.g.*, vibrational frequencies) than for direct model outputs such as energies and forces, though we present this as a motivating observation rather than a definitive criterion. Such benchmarks would provide a consistent framework for quantifying trade-offs between numerical precision, computational efficiency, and reproducibility in scientific ML research.

**Exploring high-precision models and mixed high-precision training.** Inspired by mixed-precision training (Micikevicius et al., 2018), we propose extending this concept to high-precision training by identifying precision-sensitive layers and selectively training them using FP64 arithmetic. This approach mirrors conventional mixed-precision strategies that utilize reduced precision (*e.g.*, FP32

and FP16) for most network layers while maintaining higher precision for numerically sensitive operations such as batch normalization and softmax. This direction is particularly relevant from an energy-efficiency perspective, as FP64 training is more energy-intensive than FP32. While scientific ML offers computational advantages over traditional approaches, energy consumption remains a practical concern, as we quantify in Section 5. Investigating novel model architectures and training techniques that preserve high numerical precision while enhancing energy efficiency will therefore be essential for broader adoption of scientific ML. Beyond the IEEE 754 formats considered in this work, alternative numerical representations such as *Unum* and *Posit* are a promising future direction (Gustafson & Yonemoto, 2017), though their practical evaluation currently remains limited by the lack of native hardware support.

**Collaboration with natural scientists.** Achieving meaningful progress in scientific ML requires interdisciplinary collaboration with natural scientists. This is not merely a conceptual argument but a practical requirement, as ML researchers often lack the domain-specific intuition to determine the appropriate level of numerical precision for a given scientific task. For instance, research on ML potential is published in both traditional scientific journals and ML conferences, yet the evaluation criteria and priorities differ substantially between these communities (Batatia et al., 2022; Kovács et al., 2023). Strengthening such collaboration will help bridge this gap, ensuring that precision requirements align with both scientific validity and practical usability.

**Integrating ML into traditional computational methods.** Rather than exclusively developing high-precision ML models, an alternative approach is to integrate ML into traditional computational methods to achieve both accuracy and efficiency. One promising strategy is to employ ML models while acknowledging their inherent numerical limitations and using them to generate an approximate solution (Arisaka & Li, 2023; Saverio, 2023; Napier, 2024; Kim et al., 2025). These ML-generated approximations subsequently serve as an initial guess for traditional computational methods, accelerating convergence while preserving numerical precision. Such a hybrid approach presents a compelling solution for scientific applications where both computational speed and numerical accuracy are necessary.

## 5. Alternative Views

This section presents alternative views challenging our position and offers responses to these concerns.

**Q1: Is the extra computation cost due to higher precision tolerable?** The most straightforward concern when using higher precision is the increased computational burden.

*Table 3.* Training time, energy usage, and $CO_2$ emissions when training the `MACE-OFF23-medium` model with FP32 and FP64.

| Measurement | FP32 | FP64 | Relative cost |
|---|---|---|---|
| *NVIDIA H200 GPU* | | | |
| Time (hour) | 106.8 | 134.9 | ×1.26 |
| $CO_2$ emissions (kg) | 26.70 | 38.18 | ×1.43 |
| Energy consumption (kWh) | 62.02 | 88.66 | ×1.43 |
| *NVIDIA B200 GPU* | | | |
| Time (hour) | 107.8 | 142.1 | ×1.32 |
| $CO_2$ emissions (kg) | 34.10 | 49.77 | ×1.46 |
| Energy consumption (kWh) | 79.20 | 115.59 | ×1.46 |

For example, according to NVIDIA's official specifications, H200 GPUs exhibit a 2× difference in theoretical throughput between FP64 and FP32 operations. To empirically assess this overhead, we measured the training time, carbon emissions, and energy consumption during the pretraining of the `MACE-OFF23-medium` models employed in Section 3.1, using the CodeCarbon library (Courty et al., 2024). As shown in Table 3, training in FP64 on the H200 resulted in a 1.26× increase in training time and a 1.43× increase in both energy consumption and carbon emissions.

These empirical overheads are notably lower than the theoretical 2× factor, indicating that FLOP-based estimates can be overly conservative. This is because real-world training involves substantial non-compute operations, such as data loading and transfers. While this overhead is platform-dependent, the trend persists on the NVIDIA B200, confirming that our observation is not specific to a single hardware platform. Nevertheless, it remains important to develop strategies that preserve numerical accuracy while mitigating the computational and energy costs of higher precision. Ultimately, researchers should recognize that the societal and scientific costs of incorrect conclusions due to numerical instability may far outweigh the additional computational cost of high-precision training.

**Q2: Is the issue really about numerical precision, or could it be a capacity limitation of the model?** An alternative perspective suggests that observed inaccuracies originate from fundamental limitations in network architecture or training methodologies rather than numerical precision constraints. This viewpoint posits that neural networks may lack sufficient expressivity to solve a given task, regardless of precision considerations. To distinguish numerical issues from capacity concerns, we can employ numerical analysis tools including condition numbers and numerical sensitivity analysis (*e.g.*, interval arithmetic (Hickey et al., 2001)), to determine whether errors arise from numerical instability. Since modern neural networks heavily rely on matrix operations, research on matrix sensitivity provides a analytical foundation. These insights can help clarify the relationship between numerical stability and model expressivity.

**Q3: If certain scientific computing tasks are not sensitive to numerical precision, does it matter?** While not all scientific tasks require high numerical precision, focus should be directed toward fields where high precision is essential, such as quantum chemistry and materials science, where even slight inaccuracies can lead to significant deviations. Currently, there is still limited understanding of which tasks, models, and environments are most affected by numerical precision and what factors contribute to these sensitivities. A systematic analysis is necessary to identify precision-critical cases before making broad assumptions about acceptable precision levels. Indeed, our own results include precision-insensitive cases (see Appendix B.2 and Table 5); our point is not that higher precision is always necessary. Until a clear understanding is established, a precision-aware approach should be considered, while relaxed conditions can be applied only to tasks that are demonstrably insensitive to precision.

Certain scientific tasks may not require explicit consideration of numerical precision, particularly those where logical reasoning is more critical than numerical accuracy, such as tasks relying on LLMs. These include explaining or summarizing experimental results or literature (Xie et al., 2024), generating research hypotheses (Lu et al., 2024; Kumbhar et al., 2025), providing guidance for tasks where the methodology is not clearly defined (*e.g.*, retrosynthesis), and assisting scientific education (Bewersdorff et al., 2025). In such cases, the role of ML extends beyond numerical fidelity, emphasizing knowledge synthesis and interpretability.

**Q4: Is it possible to design models that can avoid precision-related issues?** Numerical instability in scientific computing frequently originates from precision-sensitive operations including numerical differentiation, integration, and eigendecomposition. Designing scientific ML models that avoid these operations and instead directly predict their outcomes can help mitigate such instability. ML potentials exemplify this approach by directly predicting energies from atomic structures, bypassing the numerically sensitive integration and eigendecomposition required in DFT. This perspective extends to examining whether individual neural network layers are numerically stable, similar to spectral normalization (Miyato et al., 2018) which enforces Lipschitz continuity to stabilize training.

However, avoiding numerical instability through model design is not always practical, as scientists require understanding of underlying processes rather than just final predictions. This has led to ML models that mimic traditional scientific computations, such as NeuralODEs and DEQs (Chen et al., 2018; Bai et al., 2019; Wang et al., 2024), which explicitly model computational processes and align better with scientific domains emphasizing interpretability. While end-to-end approaches remain effective when predictive accuracy is the primary concern, many scientific domains continue to depend on numerical precision and computational understanding, making precision-related issues an important ongoing research area.

# 6. Conclusions

Scientific ML has become a major field in modern ML research, with the goal of developing models that contribute to scientific discovery. This position paper highlights the impact of precision issues, which can affect the practical usability of scientific ML models but have been largely overlooked. Crucially, the question of when high precision is necessary remains open; while a definitive criterion is beyond our scope, our studies tentatively suggest that sensitivity tends to arise for high-order derivatives, strong nonlinearity, and derived properties rather than direct model outputs, and our evidence is intended to motivate the systematic study of this question rather than to settle it. Beyond these methodological concerns, the precision issues in scientific ML are closely tied to ethical concerns regarding the reliability and explainability of scientific findings. In summary, our contribution lies in a practical step toward making scientific ML models more reliable, reducing the risk of misleading scientific insights due to numerical inaccuracies. If our simple yet easily actionable proposal becomes widely adopted in the field of scientific ML, it can enhance the practicality and thereby accelerate scientific discovery.

## Software and Data

All components used in our experiments are publicly available, including simulation packages, training scripts, and datasets; detailed version information, hyperparameters, and experimental settings are provided in the Appendices.

## Conflict of Interest Disclosure

The authors declare no competing interests.

## Acknowledgements

This work was supported by Institute of Information & Communications Technology Planning & Evaluation (IITP) grant funded by the Korea government (MSIT) (No. RS-2019-II190075, Artificial Intelligence Graduate School Program (KAIST)), the National Research Foundation of Korea (NRF) grant funded by the Korea government (MSIT) (No. RS-2025-00555621), the Basic Science Research Program through the National Research Foundation of Korea (NRF) funded by the Ministry of Education (No. RS-2024-00453339), and the National Supercomputing Center with supercomputing resources including technical support (No. KSC-2025-CRE-0197).

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

# A. Details of computational methods

This section provides detailed information on the DFT and FDTD calculations.

## A.1. Density functional theory calculation

**Computational environment**    Quantum mechanical calculations were performed using PySCF version 2.7.0 (Sun et al., 2018). To evaluate the impact of numerical precision, we conducted the same calculations using both single precision and double precision by declaring `np.float32` and `np.float64`, respectively. All simulations were conducted using two nodes of an AMD EPYC 7543 32-Core Processor.

**Simulation setup**    The input water molecule consists of a single oxygen atom at (0.000000, 0.000000, 0.000000) and two hydrogen atoms at (0.757000, 0.586000, 0.000000) and (-0.757000, 0.586000, 0.000000), respectively. To compare the geometry optimization result of a water molecule based on different exchange functionals, we performed both Hartree-Fock calculations and density functional theory calculations using the B3LYP functional (Becke, 1993; Yanai et al., 2004). For both methods, we employed the 6-311++G(d,p) basis set (Andersson & Uvdal, 2005). The default convergence tolerances for structural stabilization were set as follows: $|\Delta E| < 1.00 \times 10^{-6}$, RMS-Grad $< 3.00 \times 10^{-4}$, Max-Grad $< 4.50 \times 10^{-4}$, RMS-Disp $< 1.20 \times 10^{-3}$, and Max-Disp $< 1.80 \times 10^{-3}$.

## A.2. Finite-difference time-domain calculation

**Nonlinear material properties**    Kerr media were modeled with a second-order nonlinear susceptibility ($\chi^2$) ranging from $10^{-12}$ to $10^{-2}$ and the refractive index was set to 1.65 to mimic the conventional nonlinear materials like beta barium borate. The nonlinear polarization of the material was expressed as:

$$P = \epsilon_0(\chi^{(1)}E + \chi^{(2)}E^2 + \chi^{(3)}E^3 + ...)\tag{1}$$

And the second-order nonlinear polarization term is represented as: $P^{(2)} = \epsilon_0\chi^{(2)}E^2$. Meep incorporates such nonlinear polarization terms into Maxwell's equations to simulate interactions between electromagnetic waves and the material in the time domain

$$\bigtriangledown \times H = \epsilon_0\frac{\partial E}{\partial t} + \frac{\partial P}{\partial t}\tag{2}$$

$$\bigtriangledown \times E = -\mu_0\frac{\partial H}{\partial t}\tag{3}$$

**Simulation setup**    The simulation domain consisted of a 100 $\mu$m medium, a 1 $\mu$m thick boundary layer, and 2 $\mu$m buffer regions at both ends. The spatial resolution was user-defined to capture fine electromagnetic field characteristics. Kerr media were placed at the center of the domain, with $\chi^2$ explicitly defined. The calculations were performed using Meep v1.29.0 (Oskooi et al., 2010), an open-source FDTD software, with both FP32 and FP64 precisions on a single core of an AMD Ryzen 5 8500G processor.

**Source and monitor definition**    The source was defined as a Gaussian plane wave with a central wavelength of 1.55 $\mu$m and a bandwidth of 0.15. Both the source and monitors were positioned 1 $\mu$m outside the nonlinear medium, with the electric field oscillating along the x-axis. Simulations were executed to allow sufficient decay of the fields after the source was turned off to confirm accurate measurements.

**Harmonic generation and analysis**    Using the Meep's add flux function, the optical flux outside the nonlinear medium was measured, and the transmitted power spectra of the fundamental frequency ($\omega$) and harmonic components ($2\omega$, $3\omega$) were calculated. The add flux function records the time-domain values of electric and magnetic fields at specific locations, then performs a Fourier transform to convert them into the frequency domain to compute flux. This process allows precise analysis of the intensity of each frequency component within the user-defined frequency range and intervals. The analysis frequency range extended from $\omega/2$ to $3.5\omega$, encompassing all relevant frequency bands of interest. Flux measurements were particularly useful for understanding the interaction between newly generated harmonic components and existing frequency components caused by the material's nonlinearity.

**Results and reproducibility**   Simulation results demonstrated how the intensity and distribution of harmonic components varied with changes in $\chi^2$. The nonlinear modeling capabilities of Meep enabled precise analysis of harmonic generation characteristics in nonlinear optical materials.

## B. Additional experimental results

In this section, we provide additional experimental results that supplement the results presented in the main text.

### B.1. Atomic coordination difference between FP32 and FP64

A detailed examination of the atomic coordinates in Table 4 further highlights the differences. While the coordinates obtained from FP64 differ only by approximately 0.01 Å for oxygen and hydrogen atoms, FP32 results display considerable deviation. Notably, the FP32-calculated atomic positions deviate by up to 0.4 Å from those obtained using FP64, a significant difference considering that the O-H bond length itself is only 0.957 Å. In addition, the total energy difference between FP32 and FP64 calculations is approximately 1.1 Hartree (equivalent to 29.93 eV), which exceeds the formation energy of water (2.9 eV) by more than an order of magnitude. This clearly indicates that the FP32 result corresponds to a structure that cannot exist in reality. These results demonstrate that FP32 lacks the numerical precision necessary to achieve sufficient convergence tolerance in scientific computations. The failure of a simple molecular system such as water to reach an optimized structure under FP32 precision indicates its fundamental limitations in scientific calculations.

*Table 4.* Comparison of atomic coordinates and total energy for geometry-optimized water molecule at FP32 and FP64 precision levels. Both calculations used the 6-311++G(d,p) basis set. As indicated by an asterisk (*), FP32 calculations failed to converge for both HF and DFT methods, while FP64 results show compatibility between HF and DFT.

| | | **HF** 6-311++G(d,p) | | **DFT** (B3LYP) 6-311++G(d,p) | |
|---|---|---|---|---|---|
| | | FP32 | FP64 | FP32 | FP64 |
| Atomic coordinates (Å) | $O_x$ | -0.000356* | 0.000000 | 0.009524* | 0.000000 |
| | $O_y$ | 0.246311* | 0.014028 | 0.578655* | 0.000780 |
| | $O_z$ | 0.000000* | 0.000000 | 0.000000* | 0.000000 |
| | $H_1x$ | 0.453099* | 0.752792 | 0.026584* | 0.763642 |
| | $H_1y$ | 0.534244* | 0.578999 | 0.998814* | 0.585902 |
| | $H_1z$ | 0.000000* | 0.000000 | 0.000000* | 0.000000 |
| | $H_2x$ | -0.453725* | -0.752792 | -0.024889* | -0.763642 |
| | $H_2y$ | 0.534404* | 0.578999 | 0.998054* | 0.585902 |
| | $H_2z$ | 0.000000* | 0.000000 | 0.000000* | 0.000000 |
| Total energy (Ha) | | -74.938* | -76.053 | *N/A* * | -76.458 |

*Not Converged*

### B.2. Absorbed power density of a SiO$_2$ cylinder

To validate our FDTD workflow against a well-characterized linear system, we also simulated the absorption of a single silica (SiO$_2$) cylinder under normal-incidence plane-wave illumination.

**Geometry and material**   A two-dimensional square domain of side length 20 $\mu m$ was created, containing one infinitely long cylinder of radius 1.0 $\mu m$ centered at the origin. SiO$_2$ was taken from the built-in SiO$_2$ material in Meep, so its frequency-dependent permittivity—and hence refractive index—were implicitly evaluated at the simulation's center frequency. Vacuum surrounded the cylinder. Mirror symmetry along the $y$-axis halved the computational cost.

**Source definition**   A continuous-wave Gaussian plane wave, polarized along $\hat{z}$ (out-of-plane $E_z$), impinged from the left boundary. The central wavelength was 1.0 $\mu m$ with a 10 % fractional bandwidth, wide enough to sample the vicinity of the design wavelength while narrow enough to approximate monochromatic excitation.

**Monitors and post-processing**

- **Incident-flux monitor.** A line segment at $x = -2.0\mu m$ recorded the power of the incoming wave, serving as a reference for normalizing absorption.

- **Absorbed-flux box.** A closed rectangular contour wrapped tightly around the cylinder. By integrating the Poynting vector over this surface, net absorbed power $P_{\text{abs}}$ was obtained directly.

- **DFT field monitor.** A square region coincident with the flux box captured $E_z$ and $D_z$ fields in the frequency domain via Meep's discrete Fourier transform facility. Absorbed power density was then evaluated volumetrically as

$$p_{\text{abs}}(f) = 2\pi f \, \text{Im}\big(\overline{E_z}\, D_z\big) = 2\pi f \, \text{Im}(\varepsilon)\, |E_z|^2.$$

where $f$ is frequency, $\varepsilon$ is the complex permittivity of $SiO_2$, and the overbar denotes complex conjugation. Integrating $p_{\text{abs}}$ over the cylinder volume reproduced $P_{\text{abs}}$ obtained from the flux box, providing a cross-check on numerical consistency.

**Simulation parameters**  The same spatial resolution used in the nonlinear study (*i.e.* user-defined) was retained to capture sub-wavelength field variations near the curved surface. Perfectly matched layers of $2\ \mu m$ thickness enclosed the domain to eliminate spurious reflections.

This linear-dielectric benchmark served two purposes: (*i*) it confirmed the accuracy of our absorption-post-processing pipeline before applying it to nonlinear scenarios, and (*ii*) it provided a reference scale against which to compare the additional harmonic-generation pathways introduced by the Kerr media.

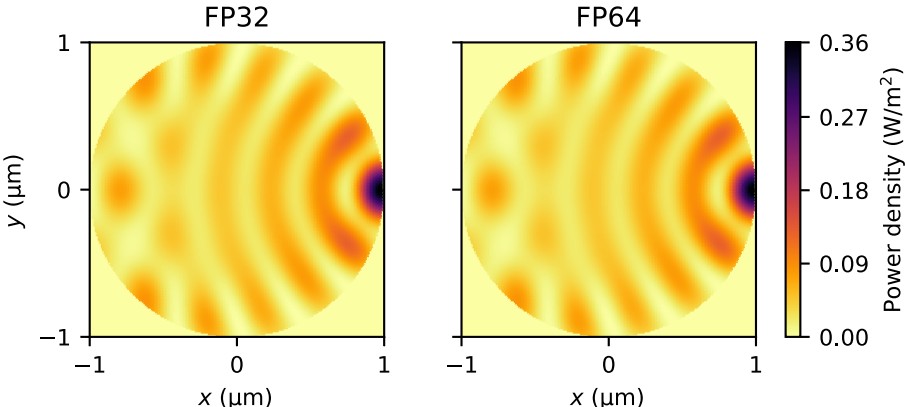

*Figure 3.* Absorbed power density of the $SiO_2$ cylinder computed with FP32 and FP64.

**Results and discussion**  In double precision (FP64) the spatially averaged absorbed–power density over the silica cylinder is $\langle p_{\text{abs}} \rangle = 3.2475 \times 10^{-2}\ (W/m^2)$, with a standard deviation $\sigma = 3.4241 \times 10^{-2}$. Using single precision (FP32) we obtained $\langle p_{\text{abs}} \rangle = 3.2476 \times 10^{-2}$ and $\sigma = 3.4243 \times 10^{-2}$. The absolute differences between the two datasets are, respectively, $\Delta\langle p_{\text{abs}} \rangle = 1.96 \times 10^{-5}$ and $\Delta\sigma = 3.02 \times 10^{-5}$, which correspond to relative errors of $6.0 \times 10^{-2}\,\%$ and $8.8 \times 10^{-2}\,\%$. These discrepancies are two orders of magnitude smaller than the intrinsic statistical spread of the fields and therefore negligible for any practical analysis.

This near-identity arises because (*i*) $SiO_2$ is essentially loss-free at the operating wavelength, so the dynamic range of $E_z$ and $D_z$ is modest; and (*ii*) the perfectly matched layers efficiently remove outgoing waves, preventing late-time reflections that could amplify numerical noise.

We therefore conclude that, for linear absorption in a dielectric cylinder, Meep's single-precision kernel yields numerically indistinguishable results from double precision while offering lower memory usage and faster runtimes. This benchmark confirms the accuracy of our FDTD workflow on a well-characterized linear system, supporting the reliability of the nonlinear Kerr-media analysis reported in the main text.

## B.3. Experiments using MACE potential

**Experimental setup**    All experiments involving MACE potentials were conducted using the following experimental setup. The MACE codebase version was 0.3.14, and the ASE library (Bahn & Jacobsen, 2002; Larsen et al., 2017) version was 3.26.0. Each experiment was performed on a system equipped with two Intel Xeon Platinum 8488C CPUs, 2 TB of RAM, and a single NVIDIA H200 GPU. Additional experiments were conducted on a separate system equipped with two Intel Xeon 6960P CPUs, 2.2 TB of RAM, and a single NVIDIA B200 GPU. Carbon emissions and energy consumption during model training were measured using the CodeCarbon library (Courty et al., 2024), version 3.2.0.

**Pretraining**    We pretrained the `MACE-OFF23-medium` model from scratch using the official implementation provided in the MACE and MACE-OFF repositories[23], together with the SPICE dataset[4] (Eastman et al., 2023) and the corresponding training scripts. The training code was modified only to integrate the CodeCarbon library for tracking carbon emissions and energy consumption. Aside from specifying the numerical precision as FP32 or FP64 in the training scripts, all other hyperparameters, including the random seed, remained unchanged from the original configuration.

*Table 5.* Comparison of energy (meV/atom) and force (meV/Å) MAE for `MACE-OFF23-medium` models trained from scratch using FP32 and FP64 precision. $\Delta$ denotes the difference between the two precision settings across various data splits.

| Data split | Metric | H200 GPU | | | B200 GPU | | |
|---|---|---|---|---|---|---|---|
| | | FP32 | FP64 | $\Delta$ | FP32 | FP64 | $\Delta$ |
| Train set | E | 0.9 | 0.8 | 0.1 | 5.1 | 5.4 | -0.3 |
| | F | 19.0 | 18.7 | 0.3 | 18.2 | 18.0 | 0.2 |
| Validation set | E | 0.9 | 0.8 | 0.1 | 5.0 | 5.3 | -0.3 |
| | F | 19.4 | 19.1 | 0.3 | 18.5 | 18.3 | 0.2 |
| DES370K Dimers | E | 0.6 | 0.6 | 0.0 | 3.6 | 3.8 | -0.2 |
| | F | 9.4 | 9.3 | 0.1 | 9.1 | 9.0 | 0.1 |
| DES370K Monomers | E | 0.6 | 0.6 | 0.0 | 4.6 | 4.6 | 0.0 |
| | F | 9.7 | 9.7 | 0.0 | 9.3 | 9.3 | 0.0 |
| Dipeptides | E | 0.6 | 0.5 | 0.1 | 4.5 | 5.0 | -0.5 |
| | F | 14.9 | 14.6 | 0.3 | 14.3 | 14.0 | 0.3 |
| PubChem | E | 1.0 | 0.9 | 0.1 | 5.8 | 6.2 | -0.4 |
| | F | 22.5 | 22.2 | 0.3 | 21.6 | 21.4 | 0.2 |
| QMugs | E | 0.8 | 0.6 | 0.2 | 2.7 | 3.0 | -0.3 |
| | F | 24.5 | 24.0 | 0.5 | 23.7 | 23.3 | 0.4 |
| Solvated Amino Acids | E | 1.4 | 1.3 | 0.1 | 8.0 | 9.4 | -1.4 |
| | F | 24.3 | 24.2 | 0.1 | 23.7 | 23.4 | 0.3 |
| Water | E | 0.8 | 0.8 | 0.0 | 2.1 | 3.9 | -1.8 |
| | F | 16.0 | 16.0 | 0.0 | 15.6 | 15.3 | 0.3 |

Table 5 reports the evaluation results of the pretrained models. Overall, the results exhibit trends consistent with those reported in the original work (Kovács et al., 2025) (Table S1). We pretrained the model on both the H200 and B200 systems under identical settings. On each platform, the difference in MAE between FP32 and FP64 remains marginal across most evaluation metrics. Notably, training on the B200 system did not complete faster than on the H200 system (Table 3), which we attribute primarily to differences in the execution environment rather than to the GPU itself: the number of CPU cores available to the training job differed between the two systems.

**Oseltamivir experiment**    We obtained the three-dimensional structure of the oseltamivir molecule from PubChem[5] and used it as the initial configuration for all experiments. All calculations were performed using the `MACE-OFF23-medium` models, together with the ASE library. Structural optimizations were carried out using the BFGS optimizer with a force tolerance of 0.005 eV/Å.

---

[2] https://github.com/ACEsuit/mace
[3] https://github.com/ACEsuit/mace-off
[4] https://www.repository.cam.ac.uk/items/d50227cd-194f-4ba4-aeb7-2643a69f025f
[5] https://pubchem.ncbi.nlm.nih.gov/compound/Oseltamivir

A comparison of the optimized oseltamivir structures obtained using the FP32 and FP64 models is shown in Figure 4. While the two structures appear visually similar, closer inspection using the black dashed guidelines reveals subtle but systematic differences in atomic coordinates. As we show next, these seemingly minor structural differences can have a significant impact on downstream vibrational properties.

For the PES analysis shown in Figure 2, we use the experimental setup illustrated in Figure 5. Starting from the optimized structures obtained separately using the FP32 and FP64 models, the oxygen atom highlighted by the dashed circle was displaced along the direction indicated by the arrow. The structures shown on the left and right correspond to the configurations obtained at the largest displacements along this path.

Vibrational frequencies computed from the optimized structures in the range of 300 to 1800 cm$^{-1}$ are reported in Table 6. Notably, multiple vibrational modes exhibit frequency differences larger than 1 cm$^{-1}$ between the FP32 and FP64 models, indicating that numerical precision can lead to non-negligible discrepancies in downstream vibrational analyses.

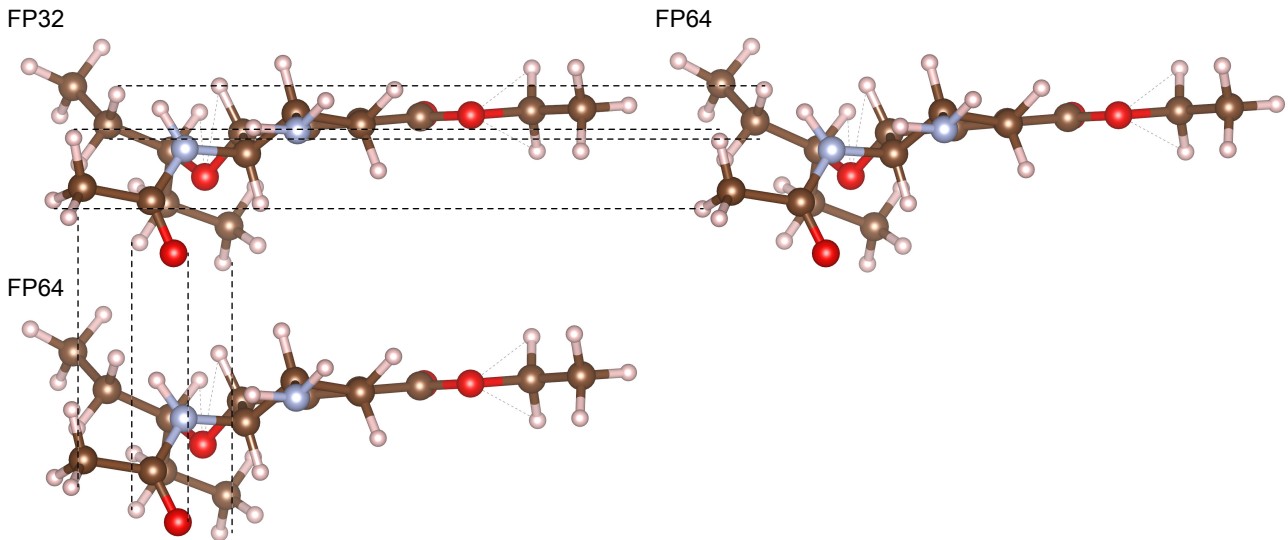

*Figure 4.* Comparison of geometry-optimized structures of the oseltamivir molecule obtained using `MACE-OFF23-medium` models trained in FP32 and FP64 precision. Black dashed guidelines highlight subtle differences in atomic positions induced by the different precision settings. These precision-induced structural variations contribute to the discrepancies in molecular properties such as vibrational frequencies reported in Tables 1 and 6.

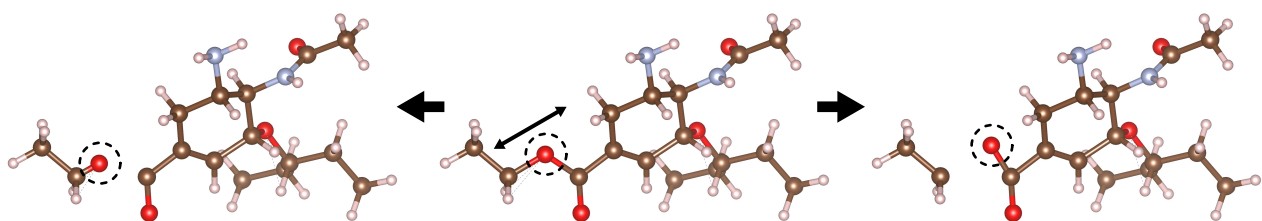

*Figure 5.* Visualization of the atomic displacement experiment performed on the oseltamivir molecule for Figure 2. Starting from the initial structure (center), a single oxygen atom indicated by the dashed circle is displaced symmetrically in opposite directions. The left and right structures correspond to configurations at equal magnitudes of displacement along the perturbation trajectory.

*Table 6.* Vibrational mode frequencies (cm$^{-1}$) of the Oseltamivir molecule computed using the `MACE-OFF23-medium` models trained in FP32 and FP64 precision. Only modes within the range of 300–1800 cm$^{-1}$ are shown out of 150 total modes. The FP32 model produces five imaginary modes, while the FP64 model produces four. $\Delta$ denotes the absolute difference between frequencies computed by the FP32 and FP64 models.

| Mode | Frequency (cm$^{-1}$) | | | Mode | Frequency (cm$^{-1}$) | | |
|------|------|------|-------|------|------|------|-------|
| | FP32 | FP64 | $\Delta$ | | FP32 | FP64 | $\Delta$ |
| 31 | 312.192 | 311.831 | 0.361 | 77 | 1155.580 | 1156.944 | 1.364 |
| 32 | 315.551 | 315.743 | 0.192 | 78 | 1168.351 | 1168.668 | 0.317 |
| 33 | 348.472 | 344.540 | 3.931 | 79 | 1176.342 | 1179.950 | 3.608 |
| 34 | 370.392 | 368.464 | 1.928 | 80 | 1181.250 | 1180.757 | 0.493 |
| 35 | 384.383 | 383.375 | 1.007 | 81 | 1200.361 | 1201.371 | 1.011 |
| 36 | 414.265 | 411.889 | 2.375 | 82 | 1221.759 | 1222.884 | 1.125 |
| 37 | 420.734 | 419.365 | 1.369 | 83 | 1250.731 | 1255.327 | 4.595 |
| 38 | 425.070 | 425.579 | 0.509 | 84 | 1258.030 | 1256.971 | 1.058 |
| 39 | 458.632 | 458.580 | 0.052 | 85 | 1268.737 | 1270.074 | 1.337 |
| 40 | 471.164 | 470.781 | 0.383 | 86 | 1278.937 | 1280.120 | 1.183 |
| 41 | 474.923 | 478.835 | 3.912 | 87 | 1294.697 | 1295.150 | 0.453 |
| 42 | 507.048 | 506.679 | 0.370 | 88 | 1299.959 | 1299.679 | 0.280 |
| 43 | 550.216 | 548.876 | 1.339 | 89 | 1305.907 | 1308.682 | 2.775 |
| 44 | 555.051 | 556.558 | 1.507 | 90 | 1312.836 | 1314.825 | 1.989 |
| 45 | 581.253 | 579.551 | 1.702 | 91 | 1329.778 | 1334.020 | 4.242 |
| 46 | 604.643 | 602.307 | 2.337 | 92 | 1339.795 | 1336.963 | 2.833 |
| 47 | 621.926 | 621.049 | 0.877 | 93 | 1341.747 | 1342.745 | 0.998 |
| 48 | 660.848 | 659.558 | 1.290 | 94 | 1363.712 | 1364.300 | 0.588 |
| 49 | 751.317 | 753.425 | 2.109 | 95 | 1376.135 | 1375.558 | 0.577 |
| 50 | 769.608 | 770.316 | 0.708 | 96 | 1393.814 | 1393.044 | 0.770 |
| 51 | 778.607 | 778.350 | 0.257 | 97 | 1396.542 | 1397.488 | 0.946 |
| 52 | 802.991 | 802.404 | 0.586 | 98 | 1398.990 | 1400.973 | 1.983 |
| 53 | 810.574 | 810.682 | 0.108 | 99 | 1402.418 | 1403.494 | 1.076 |
| 54 | 828.903 | 820.551 | 8.352 | 100 | 1411.765 | 1414.321 | 2.556 |
| 55 | 873.444 | 873.277 | 0.167 | 101 | 1414.464 | 1414.510 | 0.047 |
| 56 | 888.745 | 890.101 | 1.356 | 102 | 1420.739 | 1418.994 | 1.745 |
| 57 | 898.068 | 896.706 | 1.363 | 103 | 1426.263 | 1431.344 | 5.081 |
| 58 | 911.085 | 912.656 | 1.571 | 104 | 1429.626 | 1434.336 | 4.710 |
| 59 | 941.929 | 941.783 | 0.146 | 105 | 1438.816 | 1437.245 | 1.571 |
| 60 | 946.309 | 946.510 | 0.201 | 106 | 1470.398 | 1466.713 | 3.686 |
| 61 | 960.834 | 960.124 | 0.710 | 107 | 1471.941 | 1477.662 | 5.722 |
| 62 | 984.234 | 982.884 | 1.350 | 108 | 1478.173 | 1479.194 | 1.021 |
| 63 | 998.609 | 1001.830 | 3.221 | 109 | 1483.477 | 1479.690 | 3.787 |
| 64 | 1021.288 | 1023.270 | 1.982 | 110 | 1484.623 | 1484.687 | 0.064 |
| 65 | 1024.779 | 1028.020 | 3.240 | 111 | 1486.293 | 1487.807 | 1.515 |
| 66 | 1044.127 | 1044.189 | 0.063 | 112 | 1489.458 | 1492.576 | 3.118 |
| 67 | 1053.328 | 1051.657 | 1.671 | 113 | 1490.751 | 1493.852 | 3.101 |
| 68 | 1060.654 | 1059.565 | 1.089 | 114 | 1496.893 | 1497.904 | 1.011 |
| 69 | 1065.354 | 1064.519 | 0.836 | 115 | 1499.695 | 1502.759 | 3.063 |
| 70 | 1073.292 | 1072.305 | 0.987 | 116 | 1502.534 | 1507.718 | 5.184 |
| 71 | 1084.007 | 1084.596 | 0.588 | 117 | 1515.683 | 1516.314 | 0.631 |
| 72 | 1104.696 | 1104.967 | 0.271 | 118 | 1543.061 | 1541.125 | 1.936 |
| 73 | 1132.432 | 1129.524 | 2.908 | 119 | 1648.367 | 1646.892 | 1.475 |
| 74 | 1142.768 | 1143.646 | 0.878 | 120 | 1750.479 | 1751.317 | 0.838 |
| 75 | 1145.547 | 1146.173 | 0.625 | 121 | 1788.776 | 1789.664 | 0.887 |
| 76 | 1152.901 | 1151.876 | 1.025 | 122 | 1798.018 | 1800.110 | 2.092 |

