# OpenReview forum: "Position: Significant impact of numerical precision in scientific machine learning"
_ICML.cc/2026/Position_Paper_Track — ICML 2026 Position Paper Track regular_

### Official Review · Reviewer_FYr6 · 2026-03-07

**Significance:** 3
**Argument Clarity:** 2
**Rating:** 4
**Confidence:** 3

**Questions:**

1. The paper demonstrates precision issues through several case studies. Could the authors further clarify how general these observations are across broader scientific ML tasks?
2. The paper suggests exploring mixed high-precision training where only precision-sensitive layers use FP64. Could the authors show how such layers might be identified in practice?

**Alternative Views Section:**

Yes

**Compliance With Llm Reviewing Policy A Conservative:**

Affirmed.

**Discussion Potential:**

2

**Final Justification:**

The rebuttal addressed my concerns. I have increased my score from 3 to 4.

**Paper Summary:**

The paper states that numerical precision plays a crucial role in scientific machine learning. It provides several examples in different fields that only changing numerical precision could strongly affect the results (improve the performance). It also addressed some alternative views through experiments, which support the paper's statement.

**Position:**

Yes

**Position In Title:**

Yes

**Related Work:**

3

**Strengths And Weaknesses:**

Strengths
1. The paper is clear-written and easy to follow.
2. The experiments are conducted thoroughly, across different disciplines, and consider multiple factors.
3. The paper gathered enough evidence to support their perspective, and the discussion on counterfactual views is interesting and convincing.


Weaknesses
1. While I strongly acknowledge that most of the scientific ML scenarios mentioned in the experiments are valid (they are truly precision-sensitive), I have a doubt that, are they too specific? As the paper states that this is a problem for scientific ML, could it be better to try to generalize or summarize those precision-sensitive scenarios? There are also cases that are actually not precision-sensitive, the authors could also point that out.
2. The LLM section is weaker than previous sections in terms of evidence. Could there be more ablation on different factors that may affect the performance, as precision does (model choice, number of GPUs, batch size, etc.)? Adding at least 1 model family would help.

**Support:**

3

---

> ### Author Rebuttal · Authors · 2026-03-31
>
> We thank the reviewer for the constructive suggestion.
>
> **W1 & Q1: Generalizability**
>
> We would like to clarify our intent as a position paper.
>
> In traditional scientific computing, FP64 is the unquestioned standard (and occasionally, even quadruple precision is used for numerically sensitive problems, *e.g.*, in computational astrophysics); researchers rarely even consider FP32 as an option. However, the machine learning (ML) community has developed under a different convention where FP32 or even lower precision is widely accepted, and this gap in awareness is precisely what our paper addresses. Our manuscript is designed to span across distinct scientific domains and ML paradigms, providing concrete evidence that this gap can lead to significant errors when ML models are applied to scientific tasks.
>
> As for whether these observations can be generalized into a systematic taxonomy, we believe this is an important but currently premature goal. The scientific ML community has not yet systematically studied the relationship between precision and task characteristics, and building such a taxonomy would itself be a substantial research effort requiring community-wide investigation. Our position paper aims to motivate exactly this kind of effort.
>
> Nevertheless, based on our case studies and literature survey, we conjecture that precision sensitivity tends to increase with higher-order derivatives in the governing equations, stronger nonlinearity in the physical system, and reliance on derived quantities (*e.g.*, vibrational frequencies) rather than direct model outputs (*e.g.*, energies and forces). Validating and refining such conjectures into systematic criteria is an important direction for future work.
>
> We would also like to note that our paper does present cases where FP32 and FP64 produce nearly identical results, such as the SiO$_2$ absorption experiment in Appendix B.2 and the energy and force errors of the MACE model in Table 4. Our position is not that higher precision is always necessary, but rather that the community currently has no systematic framework to determine when it is, and this uncertainty itself is the problem.
>
> **W2: Discussion on large language models (LLMs)**
>
> We agree that Section 3.3 is more forward-looking than the other sections. There are two reasons we included this discussion. First, the massive scale of LLMs creates strong pressure to adopt low-precision representations, yet scientific applications require numerically reliable outputs, and this mismatch has not been adequately examined. Second, LLMs are increasingly applied to scientific tasks such as molecular property prediction and materials design, and we believe it is important to flag potential precision risks at this early stage. Such a forward-looking discussion is, in our view, well suited to the ICML position track.
>
> Regarding the suggestion to ablate factors such as model choice, GPU count, and batch size, Yuan et al. [1], as discussed in Section 3.3, already provides such a systematic analysis of these factors across four model families (spanning both the Qwen and Llama architectures) and a larger model (Qwen3-32B). Their key finding is that these factors all affect LLM outputs through a single root cause, namely the non-associativity of floating-point arithmetic under finite precision. We note that these factors are thus closely related to the precision issue itself, rather than being independent confounds. Our contribution in Section 3.3 is to connect these established findings to scientific applications, where numerical fidelity, not just output consistency, is the requirement.
>
> In the revision, we will clarify this distinction by framing the section explicitly as a forward-looking discussion, separate from the experimentally supported cases in Sections 2 and 3.1.
>
> **Q2: Identifying precision-sensitive layers**
>
> A promising direction would be to monitor each layer’s condition number or gradient variance during training, identifying numerically sensitive layers for selective FP64 computation. Additionally, as discussed in Section 5 (Q2), numerical analysis tools such as interval arithmetic can provide complementary information by propagating precision bounds through the network and revealing where rounding errors accumulate most.
>
> ---
>
> [1] Yuan et al., “Understanding and mitigating numerical sources of nondeterminism in LLM inference.” NeurIPS (2025).

---

> > ### Author Rebuttal · Reviewer_FYr6 · 2026-04-03
> >
> > My concerns have been resolved, I will increase my score to 4. However, I think the points raised by other reviewers are also resonable. I will keep monitoring the discussion period and make my final justification later.

---

> > > ### Author Response · Authors · 2026-04-08
> > >
> > > We sincerely thank the reviewer for the constructive feedback and for raising the score.
> > >
> > > We have addressed the concerns raised by the other reviewers during this discussion period, including a large-scale vibrational mode experiment across approximately 50,000 molecules (detailed in our response to Reviewer QJvU) and additional hardware validation on the B200 GPU (detailed in our response to Reviewer dRVJ). We hope the reviewer finds these additional results informative when making the final assessment.

---

### Official Review · Reviewer_QJvU · 2026-03-09

**Significance:** 1
**Argument Clarity:** 1
**Rating:** 1
**Confidence:** 4

**Questions:**

I would appreciate any argument against my above perspective.

**Alternative Views Section:**

Yes

**Compliance With Llm Reviewing Policy A Conservative:**

Affirmed.

**Discussion Potential:**

1

**Final Justification:**

I conclude my stand on a reject for the following reasons:
- the discussion is not nuanced enough - the main question should be when to use high precision? The most prominent case is for cases when optimization with higher order derivatives
- ML models work fundamentally differently to classical optimization. ML models perform pattern matching and given the same compute budget may deliver better results when using more parameters with a lower precision. For classical solvers this is different, where the error quite predictably increases when decreasing the compute budget.
- The presentation generally lacks more depth. This statement in the abstract has not been verified "According to several studies and our experiments, models trained with FP32 show insufficient accuracy compared to those trained with FP64, indicating that higher precision is also crucial in scientific machine learning, as in traditional scientific computing.". The chapter on PINNs is mostly about optimization instabilities. The chapter about LLMs seems somewhat unjustified.

**Paper Summary:**

The paper argues the importance of precision for scientific applications of machine learning.
The paper demonstrates 1 set of experiments and references 2 other use cases (PINNs and LLMs) where too low precision lead to worse performance.

**Position:**

Yes

**Position In Title:**

Yes

**Related Work:**

4

**Strengths And Weaknesses:**

Thank you for the effort you put into this topic and the thorough literature study.
However I disagree with the argumentation and the interpretations.
I also think the authors have a limited understanding of the difference between neural networks and classical approaches, which leads them to wrong conclusions. I'll try to build this up here.

First to the experiments:

### 2.1 and 2.2:
Agreed, it's clear that classical methods suffer from decreasing the precision. That's because their goal is to accurately compute the underlying physics.
Deep learning (DL) models are different. Their primary goal is to do pattern matching.
These 2 approaches bring completely different tradeoffs. Classical solvers quite predictably decrease the approximation error with higher precision and finer discretization.
For DL models however, given limited compute and memory budgets it might make sense to use a lower precision with a larger model (more parameters) instead. As long as it improves the model's pattern matching capabilities, this might be a very reasonable choice.
Also I argue, that DL scientists and practitioners practically work their way downwards from fp32. Before they train on half precision, they test it out on full precision. If the half precision degrades performance more than the gained compute/memory, they will stick to fp32.


### 3.1 ML Potential:
These experiments are misleading. You only compare fp64 vs fp32 but NOT to the actual ground truth (which should usually come from a classical simulation). Yes, there is a difference between fp64 and fp32, but it's not clear which model is better. You implicitly assume fp64 is better, however there is evidence that lower precision training has a regulation effect [1]. So it could be that the fp32 variant is better than fp64. These experiments are inconclusive.


2) PINNs are not a good example; they are just notoriously hard to train. While fp64 may help, PINNs seem to have more fundamental problems, such as poor conditioning [2]. While there might be something to it, there may have been unobserved/unreported confounding variables in those works. Citing 2 works with numerical issues from the ca. 22K citations of PINNs is not convincing.

3) Language models can be made deterministic with some compute overhead [3]. Also the authors somewhat leave the area of argumentation away from AI4Science towards more classical fields in AI. It is unclear why they do this.


### Call to action:
- The primary call to action "Benchmarking and reporting FP32 vs. FP64 results" misses the point. Most research transparently reports precision. Yes, in some use cases it may make sense to ablate the precision. But in principle it's the researchers' miss if they could have chosen a model with a better performance just by increasing the precision.

### Alternative views:
Q1: IMO the factors 1.26x - 1.43x from fp32 to fp64 should not be downplayed. Practical model development needs tens to hundreds of training runs. Slowing iteration time by up to 50% is a big deal. Also the additional burden on energy is not negligible.

Q2: I resonate a lot with this counter-position.

Q3: Similar argument as above. Most practitioners actually work their way down from higher to lower precision. As lower precision may induce training instabilities, this is the natural way to go.


Overall I think the authors miss the important distinctions between classical and DL based approaches.

### References
[1] https://arxiv.org/abs/2505.01043v1
[2] https://arxiv.org/abs/2402.10680
[3] https://thinkingmachines.ai/blog/defeating-nondeterminism-in-llm-inference/#conclusion

**Support:**

1

---

> ### Author Rebuttal · Authors · 2026-03-31
>
> We thank the reviewer for the detailed feedback. However, we believe these concerns stem from a misalignment regarding the paper’s scope.
>
> **1. “Deep learning (DL) models are different. Their primary goal is to do pattern matching.”**
>
> Our argument starts from the role that scientific machine learning (ML) plays in scientific research. When scientific ML is used to replace or accelerate traditional scientific computing, or more broadly to support scientific interpretation, its outputs must satisfy the reliability standards of that scientific domain, not just conventional ML evaluation criteria. From this perspective, the central question is not whether precision can sometimes be traded for efficiency in general, but whether changes in precision can alter scientifically meaningful conclusions in ways that are currently underanalyzed or underreported.
>
> **2. “ML Potential: These experiments are misleading. You only compare fp64 vs fp32 …”**
>
> The key message is that FP32 and FP64 models can lead to different scientific conclusions. In Table 1, the FP32 and FP64 models assign the observed peak at 1502 cm$^{-1}$ to different vibrational modes, leading to different physical interpretations. This precision-dependent inconsistency is itself the problem we highlight. If the same model produces different scientific conclusions depending solely on precision, researchers need to be aware of this and report it.
>
> The reviewer cites [1] as evidence that lower precision may act as regularization. However, [1] is a survey on low-precision training of large language models (LLMs), not ML potentials predicting physical quantities. The survey’s overall message is the opposite: low-precision training introduces accuracy degradation requiring mitigation techniques. The ‘regularization effect’ refers to a narrow context of binary neural networks, not scientific ML. Even if such an effect generalized to physical quantity prediction, which [1] does not show, it would strengthen our argument: the reviewer suggests that lower precision could be beneficial while also assuming practitioners can safely default to FP32, which itself underscores the need to compare and report both.
>
> **3. “Citing two works with numerical issues from the ~ 22K citations of PINNs is not convincing.”**
>
> The existence of a problem is not negated by the number of papers reporting it. As a position paper, raising awareness through such counterexamples is both appropriate and sufficient. In addition, we note that the reviewer’s own reference [2] conducts all experiments in FP64 and explicitly states “using double precision which is crucial for achieving high accuracy” in Section 4, which directly supports our position.
>
> **4. “Language models can be made deterministic” & “argumentation away from AI4Science”**
>
> The observation from [3] complements our argument. Our goal is not to claim that nondeterminism is fundamentally unavoidable, but to highlight that numerical precision can impact the reliability of LLM outputs in scientific applications.
>
> Regarding scope, we constructed Section 3.3 to be more forward-looking than other sections. LLMs face strong pressure toward low precision due to their scale. While this may be acceptable for natural language tasks, scientific tasks demand numerically reliable results. Raising awareness of this mismatch before it becomes the default in scientific applications is precisely the role a position paper should serve.
>
> **5. “Most research transparently reports precision.”**
>
> Reporting which precision was used is not the same as comparing how precision affects scientific conclusions. Such comparisons remain rare in scientific ML. Moreover, framing this as “the researchers’ miss” dismisses the possibility that the impact of precision is not yet well understood across the community.
>
> **6. On Alternative Views**
>
> We appreciate the reviewer’s engagement and find the perspectives raised by the reviewer reasonable. We plan to incorporate the reviewer’s feedback on these points in the revision, including a more thorough discussion.
>
> **7. “Authors have a limited understanding.”**
>
> We respectfully disagree. Regardless of whether scientific ML models perform pattern matching or solve physics equations, their outputs must meet the standards required by natural scientists who ultimately use them. Our experiments on the ML potential demonstrate that precision choices can alter scientific conclusions. We believe this reflects a clear understanding of both the ML and the scientific domains. We would appreciate it if the reviewer could identify specific technical errors in our analysis that justify this assessment.
>
> ---
>
> [1] Hao et al., “Low-Precision Training of Large Language Models: Methods, Challenges, and Opportunities.” preprint (2025).
>
> [2] Jnini et al., “Gauss-Newton Natural Gradient Descent for Physics-informed Computational Fluid Dynamics.” Computers & Fluids (2026).
>
> [3] thinkingmachines.ai/blog/defeating-nondeterminism-in-llm-inference

---

> > ### Author Rebuttal · Reviewer_QJvU · 2026-04-01
> >
> > I see value in raising the general point to the community that precision is important, especially in scientific ML.
> > However, IMO parts of the manuscript are not nuanced enough. Yes, precision matters, but ultimately it's the model's performance (usually measured on the test) that matters.
> > - E.g. in your abstract you say (highlighting added): "... several studies and *our experiments*, models trained with FP32 *show insufficient accuracy* compared to those trained with FP64".
> > - This is not true for your experiments as you have not compared the models to something like a test set. Even in your rebuttal you mention: "The key message is that FP32 and FP64 models can lead to different scientific conclusions."
> >
> > My suggestion would be a more nuanced discussion on when to prefer high over low precision. And then give an outlook which parts could be experimented with.
> >
> > A classical example is optimization.
> > - In the first PINN paper you cite [1] there are mostly instabilities during optimization due to the 4th derivative in the PINN: "The precision of floating-point numbers is a critical issue for the present PDE. When the calculation is executed with a
> > single precision floating-point number, the optimization is terminated due to the loss of significant digits".
> > - It makes sense for sensitive operations to use higher precision. Similar things have been done by [2] for their optimizer. There is also a pytorch guide on which operations are float16-able [3].
> > - also the second PINN paper [4] mostly just uses higher precision for a more stable optimization
> >
> >
> > As a corollary, I think the scope is too wide with the LLM example.
> > - the LLM example detour does not really fit. If you want to show a broader statement, consider renaming or restructuring that subsection. LLMs are trained with rigorous performance testing, like "can we save compute while keeping the loss at the same level?".
> >
> >
> >
> > [1]: Nakamura, Y., Shiratori, S., Takagi, R., Sutoh, M., Sugihara, I., Nagano, H., and Shimano, K. Physics-informed neural network applied to surface-tension-driven liquid film flows
> >
> > [2]: Dangel, F., Müller, J. and Zeinhofer, M., 2024. Kronecker-factored approximate curvature for physics-informed neural networks. Advances in Neural Information Processing Systems, 37, pp.34582-34636.
> >
> > [3]: https://docs.pytorch.org/docs/stable/amp.html#cuda-ops-that-can-autocast-to-float16
> >
> > [4]: Sharma, R. and Shankar, V., 2022. Accelerated training of physics-informed neural networks (pinns) using meshless discretizations. Advances in neural information processing systems, 35, pp.1034-1046.

---

> > > ### Author Response · Authors · 2026-04-08
> > >
> > > We thank the reviewer for the continued discussion. We address each point below.
> > >
> > > **Abstract revision**
> > >
> > > We acknowledge this point. In the revision, we will revise the abstract to more accurately reflect our findings, *e.g.*, “FP32 and FP64 models can yield different scientific conclusions, and this discrepancy is currently underreported.”
> > >
> > > **Large-scale vibrational mode experiment**
> > >
> > > Motivated by the reviewers’ feedback, we conducted additional experiments to provide stronger quantitative support for our position. We conducted a large-scale vibrational mode analysis across the MACE-OFF23 test set [1], which contains 50,195 molecules, of which 49,804 were successfully computed. For each molecule, we performed structure optimization and vibrational frequency calculation using the FP32 and FP64 model checkpoints trained on the H200 GPU as described in Section 3.1. The calculation settings were fmax = 0.005 eV/Å for structure optimization convergence, finite displacement δ = 0.01 Å, and nfree = 4 (which is stricter than the nfree = 2 setting used in Section 3.1). All calculations were performed on a single NVIDIA B200 GPU, as our computational resources transitioned from H200 to B200 during the discussion period.
> > >
> > > We then identified mode misalignment, defined as cases where the FP64 mode $i$ with frequency $f_{i}^{FP64}$ and the nearest-frequency FP32 mode $j$ with frequency $f_{j}^{FP32}$ satisfy $i \neq j$ and $|f_{i}^{FP64} - f_{j}^{FP32}|<1$ cm$^{-1}$. As explained in our manuscript, this “misalignment” could be crucial for real-world scientists, as it may lead to incorrect vibrational mode assignments in spectral analysis. To minimize false positives from near-degenerate modes, we excluded cases where consecutive FP64 modes are within 0.5 cm$^{-1}$ of each other. These criteria are deliberately conservative, meaning the actual number of precision-dependent discrepancies is likely higher.
> > >
> > > The results are as follows. Out of 49,804 molecules, 42,406 (85.15%) exhibit at least one mode reordering, with an average of 5.13 reorderings per molecule. Furthermore, the prevalence of reorderings increases systematically with molecular size:
> > >
> > > | Atom count | Molecules | With reorderings | Percentage | Total reorderings |
> > > |:----------:|:---------:|:----------------:|:----------:|:-----------------:|
> > > |    1-10    |   3,824   |       1,285      |    33.6    |       2,208       |
> > > |    11-20   |   10,882  |       7,121      |    65.4    |       17,121      |
> > > |    21-50   |   34,784  |      33,686      |    96.8    |      228,616      |
> > > |   51-100   |    312    |        312       |    100.0   |       7,273       |
> > > |    100+    |     2     |         2        |    100.0   |        415        |
> > >
> > > This trend is physically expected: larger molecules have denser vibrational spectra, making mode ordering more susceptible to small numerical perturbations introduced by different precision settings.
> > >
> > > We emphasize that the point of this experiment is not to determine which precision is closer to ground truth. Rather, it demonstrates that precision choice alone can alter vibrational mode assignments across the vast majority of molecules in a standard benchmark. Since mode misalignment directly determines spectral peak interpretation in experimental research, this systematic inconsistency is precisely the kind of underreported issue that our position paper aims to highlight. We will include these results in the camera-ready manuscript.
> > >
> > > **On PINNs**
> > >
> > > We appreciate the reviewer’s nuanced perspective on PINNs. We agree that the relationship between numerical precision and PINNs deserves a more careful treatment than our current manuscript provides, and we will refine our discussion accordingly in the revision.
> > >
> > > **On the LLM section scope**
> > >
> > > Multiple reviewers (Reviewers cmCn, FYr6, and QJvU) have noted that Section 3.3 is less established than the other sections. We agree with this assessment. In the revision, we will restructure Section 3.3 to clearly position it as a forward-looking discussion rather than established experimental evidence, and explicitly distinguish it from the experimentally supported sections (Sections 2 and 3.1).
> > >
> > > ---
> > >
> > > [1] https://doi.org/10.17863/CAM.107498

---

### Official Review · Reviewer_cmCn · 2026-03-11

**Significance:** 2
**Argument Clarity:** 3
**Rating:** 4
**Confidence:** 2

**Questions:**

1. In ML settings, when is FP64 really necessary, and when is FP32 likely enough? Are there practical signs that a task is likely to be precision-sensitive?

2. In several examples, FP32 performs worse than FP64. How much of that gap comes from numerical precision itself? as opposed to other factors such as optimization difficulty, implementation details?

3. The author(s) call to action of reporting FP32/FP64 comparisons and, when appropriate, releasing FP64 models.
Based on my experience, though, many researchers do not have the budget (time or computing resources, to name a few) to do this systematically.
What would the authors consider a reasonable minimum standard here? For example, would targeted checks on the most sensitive tasks or components already be useful?

4. Compared with the sections on DFT, FDTD, ML potentials, the discussion of LLM-based scientific applications seems more forward-looking. Is the discussion of LLM as current evidence for the main position, or more as a promising future direction?

**Alternative Views Section:**

Yes

**Compliance With Llm Reviewing Policy A Conservative:**

Affirmed.

**Discussion Potential:**

3

**Final Justification:**

I still don't completely agree with the authors' view, but the position they raised is worth discussing and further experimentally verifying, and that's also a contribution to the community.

Therefore, I'm raising my score.

**Paper Summary:**

The paper argues that numerical precision is an important and still underappreciated issue in scientific machine learning. The main point is that in some scientific applications, small numerical errors are not just minor implementation details, but can affect physically meaningful conclusions. The authors support this claim with examples from both traditional scientific computing and several scientific ML settings, such as DFT, FDTD, ML potentials, PINNs, and LLM-based scientific tasks. Based on this discussion, the paper encourages the community to report precision settings more explicitly and to pay closer attention to FP32/FP64 differences when evaluating scientific ML systems.

**Position:**

Yes

**Position In Title:**

Yes

**Related Work:**

3

**Strengths And Weaknesses:**

**Strengths:**
1. The paper makes a clear case that numerical precision deserves more attention in scientific machine learning, especially in applications where small numerical differences can affect scientifically meaningful results. In my opinion, this is a relevant topic for the ICML community, as scientific ML is increasingly used in settings where reliability and reproducibility matter.
2. The paper does not make the argument only at a high level, but discusses examples from traditional scientific computing as well as several scientific ML areas, including ML potentials, PINNs, and LLM-based scientific applications.
3. The ends of the paper with practical recommendations, such as reporting precision settings more clearly and checking FP32/FP64 differences when they may matter.

**Weakness:**
1. The discussion of LLM-based applications feels less established than the sections on more classical scientific ML problems.

2. The paper would benefit from being more explicit about which claims are already well supported.

**Support:**

2

---

> ### Author Rebuttal · Authors · 2026-03-31
>
> We thank the reviewer for the constructive feedback. Before addressing individual points, we note that the core contribution of this paper is not to provide complete empirical answers, but to be the first to unify precision-related issues across multiple scientific machine learning (ML) domains into a single structured problem for the community to address. We believe this framing is important for interpreting our responses below.
>
> **W1 & Q4: Discussion on large language models (LLMs)**
>
> We agree that Section 3.3 (LLM section) is more forward-looking than the other sections, and we included it for two reasons:
>
> 1. LLMs face strong pressure toward low precision due to their scale, and while this is acceptable for natural language tasks where output diversity is tolerable, scientific tasks demand numerically reliable results, making this a concerning mismatch.
> 2. LLM-based scientific research is growing rapidly, and we believe it is important to raise awareness of this issue across the community before low-precision practices become the default in scientific applications. This is precisely the role a position paper can serve.
>
> In the revision, we will clarify the role of Section 3.3 as a forward-looking discussion and distinguish it more clearly from the other sections.
>
> **W2: Clarity on claim support levels**
>
> Based on the reviewer’s comment, we will revise the manuscript to clearly distinguish (1) established facts from prior literature, (2) new evidence from our experiments, and (3) open questions requiring future work.
>
> **Q1: When is FP64 necessary?**
>
> We agree that this is a central question for the area, and we do not claim that this paper provides a complete criterion for when FP64 is necessary. Rather, one purpose of this position paper is to show that this question is scientifically important, application-dependent, and still insufficiently studied across the scientific ML domain. From our case studies and literature survey, we observed that precision sensitivity tends to increase with higher-order derivatives in differential equations, stronger nonlinearity in the physical system, and the use of derived properties (*e.g.*, vibrational frequencies) rather than direct model outputs (*e.g.*, energies and forces). We present these as motivating observations rather than as a definitive rule, and developing them into systematic criteria is an important open problem that this position paper aims to highlight.
>
> **Q2: Precision vs. other factors**
>
> Our MACE experiments in Section 3.1 were designed to isolate the effect of numerical precision from other factors such as model capacity or optimization difficulty. As detailed in Appendix B.3, we used the official MACE repository with the same codebase and all hyperparameters, including the random seed, remained the same across runs, and only the numerical precision was changed. Therefore, any observed differences between FP32 and FP64 models are most directly explained by the difference in numerical precision, as all other experimental conditions were held constant. However, for the broader question of how numerical precision interacts with other factors across different models and tasks, systematic investigation is still needed, which is one of the research directions we advocate.
>
> **Q3: Reasonable minimum standard**
>
> We agree that conducting full FP32/FP64 retraining is not always feasible for all researchers. As we discuss in our manuscript, since our results show that direct outputs are relatively robust while derived properties are more sensitive, we suggest that researchers focus precision checks on the most sensitive downstream tasks in their application.  At the community level, sharing FP64 checkpoints for representative models would further reduce individual burden. For instance, in ML potentials, once the best hyperparameters are identified in one precision setting, conducting a single additional training run in the other precision and comparing derived properties such as phonon spectra or vibrational frequencies would serve as a practical minimum standard that can be briefly reported in an appendix.

---

> > ### Author Rebuttal · Reviewer_cmCn · 2026-04-01
> >
> > Q1 and Q2 remain insufficiently resolved, making the discussion of Q3 and the author's call to action seem less compelling at its current stage. The experiment in 3.1 may not be sufficiently representative. Based on common empirical observations, model performance is not strongly correlated with numerical precision formats such as FP64, FP32, or even FP16 and mixed precision. Therefore, I think reporting FP64 results could be further clarified.

---

> > > ### Author Response · Authors · 2026-04-08
> > >
> > > We thank the reviewer for the continued engagement.
> > >
> > > **Q1: When is FP64 necessary?**
> > >
> > > As we noted in our first response, providing a complete criterion is beyond the scope of this paper. Nevertheless, in response to the reviewer’s request for broader evidence, we conducted a large-scale vibrational mode analysis across the MACE-OFF23 test set to provide stronger empirical grounding beyond the single-molecule case study in Section 3.1. For each molecule, we performed structure optimization and vibrational frequency calculation using both FP32 and FP64 model checkpoints with identical settings. The detailed criteria for identifying mode reorderings and the experimental setup are provided in our response to Reviewer QJvU.
> > >
> > > The results show that primary metrics such as energy and force MAE (Table 4) exhibit negligible differences between FP32 and FP64, which is consistent with the reviewer’s observation that model performance is generally not strongly correlated with numerical precision format. However, the results for derived properties show a significantly different trend. Out of 49,804 molecules, 42,406 (85.15 %) exhibit at least one vibrational mode misalignment, ranging from 33.6 % for molecules with 1-10 atoms to 100 % for molecules with 51+ atoms. This indicates that precision sensitivity depends on whether the quantity of interest is a direct model output or a higher-order derived property, and that primary metrics alone cannot determine whether a given task is precision-sensitive.
> > >
> > > **Q2: Isolating precision effects**
> > >
> > > As detailed in Appendix B.3, our MACE experiments use the same codebase, hyperparameters, and random seed across runs, with only the numerical precision changed. The large-scale experiment above follows the same protocol. Therefore, the observed discrepancies are attributable to the change in numerical precision under otherwise matched experimental conditions, rather than to changes in model architecture, codebase, or hyperparameter settings.
> > >
> > > **Q3: Practical relevance of reporting FP64 results**
> > >
> > > The reviewer noted that model performance is generally not strongly correlated with numerical precision format. However, UMA [1], a recent large-scale machine learning interatomic potential, adopted BF16 for pretraining but found it necessary to fine-tune in FP32 to recover accuracy, demonstrating that precision choices can materially affect model performance even in state-of-the-art systems. Notably, their analysis stops at FP32 and does not examine FP64 or its impact on derived properties. Given our finding that 85 % of molecules exhibit mode reorderings between FP32 and FP64, the precision sensitivity observed in UMA may extend further than their current evaluation reveals, which is precisely the kind of gap that systematic precision reporting would help identify.
> > >
> > > ---
> > >
> > > [1] Wood et al., “UMA: A Family of Universal Models for Atoms.” NeurIPS (2025).

---

### Official Review · Reviewer_dRVJ · 2026-03-13

**Significance:** 3
**Argument Clarity:** 4
**Rating:** 4
**Confidence:** 5

**Questions:**

1. In table 2 shows the runtime and energy cost between FP64 and FP32. My view is that this argument is weak. If the computational platform is optimized for FP64, the savings when running FP32 will be small. An extreme case is that if a computing platform is optimized for FP16, critical micro architecture decisions such as floating point path width and number of FPU units, load/store bandwidth, would make running FP64 extremely inefficiently. Therefore, blindly running experiments on existing hardware platform may not provide sufficient insight to the pros and cons of a decision on certain numerical precision.

2. Closely related to 1, there are also other FP representation formats other than IEEE 754, such as Unum. It would be ideal to include them as well.

**Alternative Views Section:**

Yes

**Compliance With Llm Reviewing Policy A Conservative:**

Affirmed.

**Discussion Potential:**

4

**Paper Summary:**

This position paper raise the need of higher numerical precision for scientific machine learning applications. More specifically whether higher numerical precision, FP64, is needed when training models for scientific machine learning applications. The position is both defensible and falsifiable

**Position:**

Yes

**Position In Title:**

Yes

**Related Work:**

3

**Strengths And Weaknesses:**

**Strength**

The position is timely and significant, especially when scientific machine learning is gaining traction. The authors did a good job explaining the background and also provided sufficient empirical evidence that the position expose a potential a knowledge gap which needs to be address.

The paper also provide some argument to illustrate the defendability of the position. The alternative views section provide sufficient falsifiability to the position, although I believe a few components are missing. The references is sufficiently comprehensive.

**Weakness**
The position paper assumes the hardware implementation is _static_, or in other words, based on the ML hardware platforms currently available on market. In reality, all hardware designs are results of optimization (e.g., optimized for certain workloads). See the questions section.

**Support:**

3

---

> ### Author Rebuttal · Authors · 2026-03-31
>
> We thank the reviewer for the thoughtful and constructive evaluation.
>
> **Clarification on scope**
>
> Our position is not simply that high precision, such as FP64, should always be used. Rather, we argue that the differences between FP32 and FP64 should be systematically analyzed and reported in scientific machine learning (ML) research, as stated in our position statement (Section 1) and core recommendation (Section 4).
>
> **Q1: Hardware platform dependency**
>
> We understand the reviewer’s concern regarding the platform dependency of Table 2 and agree that the overhead could vary across differently optimized hardware platforms. Nevertheless, the H200 GPU we used in our experiments represents a practically relevant platform in today’s high-performance computing landscape, where GPU-accelerated systems are widely adopted (*e.g.*, as reflected in the TOP500 list [1]), and our results therefore provide a meaningful and representative baseline. Moreover, on FP64-optimized hardware, the cost gap would likely be further reduced, which is consistent with our broader argument.
>
> However, we would like to emphasize that Table 2 addresses only one aspect of our argument, the computational cost concern. The more fundamental issue is correctness: Hardware optimization does not close the intrinsic precision gap between FP32 and FP64. The convergence failures in density functional theory calculations and vibrational mode misassignments that we report would occur regardless of the hardware platform, because they stem from insufficient numerical resolution, not from computational inefficiency. From the perspective of natural scientists, theory-based computations are inherently expected to yield physically reliable results. In fact, acceleration techniques are already widely accepted, but only when they are mathematically guaranteed to preserve numerical fidelity. A computation that produces scientifically imprecise results is simply not useful, regardless of how fast it runs.
>
> In the revised manuscript, we will explicitly note the platform dependency of Table 2 and clarify the distinction between computational cost and numerical correctness.
>
> **Q2: Alternative representations such as Unum**
>
> We limit our scope to IEEE 754 as it is the *de facto* standard supported by almost all current hardware and ML frameworks. Regarding Unum and Posit, these are theoretically interesting alternatives, but currently available GPU hardware does not natively support these formats, making them difficult to adopt or evaluate in practice for both the authors and the broader scientific computing community. We will mention them as a future direction in the revised manuscript.
>
> ---
>
> [1] https://top500.org/lists/top500

---

> > ### Author Rebuttal · Reviewer_dRVJ · 2026-04-02
> >
> > I can accept the argument on alternative data formats such as Unum. However, I need more argument on Table 2, in the sense that HW designers have a lot of freedom to optimize the HW based on a given set of workloads. Using one HW platform may not be representative. One solution is to use the HW from the same vendor, but with different configuration to provide a better picture. Different platforms from NVIDIA (but in the same generation of technology) would be a good start.

---

> > > ### Author Response · Authors · 2026-04-08
> > >
> > > We thank the reviewer for the continued engagement and the constructive suggestion.
> > >
> > > **Hardware dependency of Table 2**
> > >
> > > We acknowledge the reviewer’s point that a single hardware platform may not be fully representative for cost comparisons. We would have preferred to test on a different configuration within the same Hopper generation, as the reviewer suggested, but our available resources limited us to the NVIDIA B200 (Blackwell architecture). Although this differs from the H200 in generation rather than configuration, we believe it nonetheless provides a meaningful cross-hardware check of whether our main observation persists beyond the original platform. Training the MACE-OFF23 model from scratch on the B200 is feasible, but the remaining discussion period does not allow sufficient time to complete the full training run. We will complete this experiment and incorporate the cost comparison results into the camera-ready version.
> > >
> > > **Correctness across hardware platforms**
> > >
> > > In addition to the cost analysis, we note that the core correctness concern does not appear to be specific to the original hardware platform. To demonstrate this, we transferred the FP32 and FP64 model checkpoints used in Section 3.1 (originally trained on the H200 GPU) and performed the oseltamivir vibrational mode experiment (Table 1) on the B200 GPU. The results reproduce the same trend: the two models yield different vibrational mode orderings near 1502 cm$^{-1}$, leading to different physical interpretations of the same spectral region. This suggests that the precision dependent discrepancy is primarily attributable to the model weights rather than hardware specific inference behavior, reaffirming that numerical fidelity is a prerequisite for scientifically meaningful interpretations.
> > >
> > > |      | Mode | FP32 freq. (cm$^{-1}$) | FP64 freq. (cm$^{-1}$) | Diff. |
> > > |:----:|:----:|:----------------------:|:----------------------:|:-----:|
> > > | H200 |  114 |         1496.89        |         1497.90        |  1.01 |
> > > |      |  115 |         1499.69        |       **1502.75**      |  3.06 |
> > > |      |  116 |       **1502.53**      |         1507.71        |  5.18 |
> > > | B200 |  114 |         1496.85        |         1497.90        |  1.05 |
> > > |      |  115 |         1499.74        |       **1502.75**      |  3.01 |
> > > |      |  116 |       **1502.58**      |         1507.71        |  5.13 |
> > >
> > > Furthermore, we extended this analysis across the MACE-OFF23 test set (49,804 molecules), where 85.15 % of molecules exhibit at least one mode reordering between FP32 and FP64. This hardware-independent reproducibility further reinforces our argument: the precision-dependent discrepancies originate from the numerical representation itself, not from platform-specific artifacts. The detailed results and experimental setup are provided in our response to Reviewer QJvU.

---

### Decision · Program_Chairs · 2026-04-30

**Decision:**

Accept (regular)

**Comment:**

Reviewers mostly agree that this paper presents a position that will be of interest to the ICML community insofar as most recent efforts have been directed toward lower precision (e.g. FP16, FP8) for efficiency gains rather than increased precision. While the paper itself argues explicitly for higher precision in the form of FP64, a recurring question from reviewers that could ideally be addressed more directly (or explicitly relegated to a topic out of scope) is *when* we should be using various FP8/16/32/64 precision formats. This consideration will likely be top-of-mind for many readers. Nevertheless, this position might reflect and entry point that sparks this discussion among the broader community.

There are several substantive concerns regarding the depth of discussion that appears in this position. Unsurprisingly, there is more depth provided in the discussion of classical numerical methods, which makes sense given the places that additional precision stands to make a difference. However, the result in presentation is that the discussion of LLM-based scientific approaches reads as comparatively weakly supported. This is partially addressed in rebuttal but would be important to improve in revision. Another major concern relates to the isolation of precision in the experimental design. There are meaningful changes suggested in the rebuttal (e.g. validation on additional hardware types), but there are still likely to be concerns about other factors that could be normalized. For example, given a fixed time such as the amount of time allowed for FP32 computation, is there a benefit to FP64 training. While not necessary to include, these are likely questions that are alluded to in review that will likely be salient for a broader audience.

Overall, despite these concerns, this paper could spark meaningful discussion for ICML that motivates future work in the space.